# LANGPROP: A CODE OPTIMIZATION FRAMEWORK USING LANGUAGE MODELS APPLIED TO DRIVING

## ABSTRACT

LangProp is a framework for iteratively optimizing code generated by large language models (LLMs) in a supervised/reinforcement learning setting. While LLMs can generate sensible solutions zero-shot, the solutions are often suboptimal. Especially for code generation tasks, it is likely that the initial code will fail on certain edge cases. LangProp automatically evaluates the code performance on a dataset of input-output pairs, as well as catches any exceptions, and feeds the results back to the LLM in the training loop, so that the LLM can iteratively improve the code it generates. By adopting a metric- and data-driven training paradigm for this code optimization procedure, one could easily adapt findings from traditional machine learning techniques such as imitation learning, DAgger, and reinforcement learning. We demonstrate the first proof of concept of automated code optimization for autonomous driving in CARLA, showing that LangProp can generate interpretable and transparent driving policies that can be verified and improved in a metric- and data-driven way. Our code will be open-sourced and is available at `https://github.com/langprop-iclr24/LangProp`.

## 1 INTRODUCTION

Building systems that can self-improve with data is at the core of the machine learning paradigm. By leveraging vast amounts of data and having an automated feedback loop to update models according to an objective function, machine learning methods can directly optimize the metrics of interest, thus outperforming systems that are handcrafted by experts. In the early history of artificial intelligence (AI), Symbolic AI, e.g. rule-based expert systems (Hayes-Roth, 1985; Jackson, 1986), was a dominant and perhaps a more intuitive and explainable approach to solving tasks in an automated way, and is still widely used in fields such as medicine (Abu-Nasser, 2017) and autonomous driving (Badue et al., 2021). However, there have been numerous successes in recent decades in machine learning, e.g. deep neural networks, that demonstrate the advantage of data-driven learning.

The innovation in Large Language Models (LLMs) (Brown et al., 2020; OpenAI, 2023; Touvron et al., 2023) is a prominent success enabled by neural networks. Trained on both natural language and code, they can translate human intent and logic into executable code and back, expanding the boundaries of applying logic and reasoning. Unlike other machine learning techniques, LLMs have an affinity with Symbolic AI since they operate in discrete symbolic input-output spaces. The generated outputs are interpretable, even though the internal representation of these tokens is in a continuous embedding space. This observation led us to question if it is possible to have the best of both worlds – having an interpretable and transparent system, characteristic of Symbolic AI, which can self-improve in a data-driven manner, following the machine learning paradigm. We believe that LLMs provide the missing piece of the puzzle; the optimization mechanism.

Our insight is that we can draw a direct analogy from training neural networks and *train* symbolic systems by leveraging the power of language models to interpret and generate scripts. Using the analogy of model training, an LLM can be used as an *optimizer* equivalent to stochastic gradient descent or Adam. The actual *model* in our paradigm is an object that handles the initialization and updates of *parameters* as well as the forward pass logic, where the *parameters* are a collection of symbolic scripts that the LLM generates. At every iteration, we perform a forward pass through the model, compare it against the ground truth in the dataset, and pass the scores and feedback into the LLM which interprets the results and updates the scripts in a way that fixes the issues raised.

While many methods use LLMs for code generation, and systems such as Auto-GPT (Richards, 2023) iteratively query LLMs to execute tasks in an agent-like manner, as far as we know, we are the first to completely translate and apply the training paradigm used in machine learning for iterative code generation. We draw inspiration from MineDojo VOYAGER (Wang et al., 2023), which first introduced the idea that a collection of code generated by LLMs (skill library) can be considered as sharable and fine-tunable *checkpoints*. However, VOYAGER's implementation is specific to Minecraft, and additional work is needed to apply its approach to other domains. We propose Lang-Prop, a general code optimization framework that is easily adaptable to many application domains.

Autonomous driving is a key area in which model interpretability and transparency are critical. We consider LangProp to be a valuable proof of concept for building interpretable and language-instructable systems in a more automated and learnable way. We tested our hypotheses that (a) LangProp can generate interpretable code that learns to control a vehicle, (b) LangProp can improve driving performance with more training data in comparison to zero-shot code generation, and (c) we can easily transfer training paradigms from machine learning to LangProp such as imitation learning, reinforcement learning (Sutton & Barto, 2018) and DAgger (Ross et al., 2011).

## 2 RELATED WORK

### 2.1 LLMS FOR CODE GENERATION

Transformer-based models (Vaswani et al., 2017) have shown outstanding performance in code generation tasks (Chen et al., 2021; Li et al., 2022; Xu et al., 2022; Nijkamp et al., 2023; Fried et al., 2023). In particular, general purpose LLMs (Ouyang et al., 2022; OpenAI, 2023) have shown remarkable capabilities of code generation, translating natural language into code, and vice versa. However, there is no guarantee that the generated code is error-free. Benchmarks have been suggested to evaluate LLMs on the quality of code generation (Chen et al., 2021; Liu et al., 2023).

Code generation with execution is especially relevant to our work. Cobbe et al. (2021) and Li et al. (2022) used majority voting on the execution results to select code from a pool of candidates. but this is prone to favoring common erroneous solutions over infrequent correct solutions. Ni et al. (2023) suggested a ranking mechanism using a learned verifier to assess code correctness. Given the code, its specification, and its execution results, it computes the rankings based on the code correctness and code generation probability. CLAIRIFY (Skreta et al., 2023) implemented automatic iterative prompting that catches errors and provides feedback to the LLM until all issues are resolved.

Tangentially related fields are Automated Program Repair (APR) (Xia & Zhang, 2022; Xia et al., 2022), unit test generation (Roziere et al., 2022), and planning applied to LLMs and code generation (Le et al., 2022; Zhang et al., 2023). APR is typically solved as a text infill task by identifying an erroneous block of code, masking it out, and querying an LLM, providing the surrounding code as context. Planning for LLMs formulates code generation as a sequence generation task and applies Reinforcement Learning techniques. While these approaches are orthogonal to our approach of iteratively generating code using a pre-trained general-purpose LLM as an optimizer, findings from these fields may be compatible with LangProp for future work.

### 2.2 LARGE LANGUAGE MODELS FOR AUTOMATING COMPOSITIONAL TASKS

LLM-powered agents have demonstrated sophisticated planning capabilities. Sequential prompting with the history of observation, action, and the reason for the action was proposed by ReAct (Yao et al., 2023) as an improvement to Chain-of-Thought prompting (Wei et al., 2022), which has also been applied to autonomous driving Fu et al. (2023). Auto-GPT (Richards, 2023) automated tasks by iteratively generating a sequence of subtasks in finer detail until they are executable. A similar strategy was applied to robotics (Huang et al., 2022). SayCan (Ahn et al., 2022) used LLMs to generate candidate subgoals and assessed their affordances with a value function given visual observations to ground the agent's behavior. VIMA (Jiang et al., 2023) and PaLM-E (Driess et al., 2023) demonstrated profound reasoning and execution capabilities on multi-modal tasks such as Visual Q&A and robotics by fine-tuning LLMs to allow multi-modal prompting. Inner Monologue (Huang et al., 2023) used environment and user feedback to replan for embodied tasks. Unlike our method, the above methods require an LLM in the loop during inference, whereas our method only requires access to an LLM during the code optimization stage. Liang et al. (2023) and Singh et al. (2023)

used LLMs to directly generate code for robotics, while ViperGPT (Dídac et al., 2023) and Vis-Prog (Gupta & Kembhavi, 2023) composed pre-trained vision-and-language models to solve challenging vision tasks which require reasoning and domain knowledge. However, none of the above methods implement code optimization via iterative prompting.

Our method is inspired by VOYAGER (Wang et al., 2023), which integrates environment feedback, execution errors, and self-verification into an iterative prompting mechanism for embodied control in Minecraft. VOYAGER maintains a *skill library*, a collection of verified reusable code, which can be considered as *checkpoints*. However, there is no mechanism to optimize or remove a sub-optimal skill once it has been added to the library. We address this limitation and present a more general code optimization framework that can be applied to a variety of domains, e.g. autonomous driving.

### 2.3 AUTONOMOUS DRIVING AND THE CARLA BENCHMARK

Approaches to Autonomous Driving can be broadly classified into modular systems and end-to-end systems (Yurtsever et al., 2020). Most systems take a modular approach (Urmson et al., 2008; Levinson et al., 2011; Wei et al., 2013; Maddern et al., 2017), which has human-defined rules that orchestrate separately engineered components for localization and mapping, object detection, tracking, behavior prediction, planning, and vehicle control. Such systems allow compartmentalization and better interpretability, but can be complex and require domain knowledge to maintain and update. Another challenge is error propagation (McAllister et al., 2017), i.e. the upstream outputs can be erroneous and must be corrected downstream. Recent work has harnessed end-to-end learning to address these issues. Imitation learning (IL) (Bojarski et al., 2016; Bansal et al., 2018) optimizes the policy to match actions taken by experts, and is the most widely used approach. However, its performance is upper-bounded by the expert. Deep reinforcement learning has also shown successes in simulation (Sallab et al., 2017), on the road (Kendall et al., 2019), and in combination with IL (Lu et al., 2022). Our work combines both the benefit of interpretability of expert systems while also taking a data-driven approach, exposing the system to potential failure modes and adverse scenarios during training time and iteratively optimizing the system towards a well-defined driving metric so that the resulting system is robust to adverse events and potential errors in intermediate components.

CARLA (Dosovitskiy et al., 2017) is a widely used open-sourced 3D simulator for autonomous driving research. Many prior works on CARLA have open-sourced their expert agents. Roach (Zhang et al., 2021) trained a PPO agent (Schulman et al., 2017) on handcrafted reward signals with privileged information. The heavy lifting is done at the reward shaping level, where hazardous agents are identified and the desired speed and pose are computed. Roach expert is also used in MILE (Hu et al., 2022) and TCP (Wu et al., 2022), where TCP has an additional emergency braking upon detecting potential collisions. TransFuser (Chitta et al., 2022), InterFuser (Shao et al., 2023) and TF++ (Jaeger et al., 2023) implement their handcrafted expert systems, either using cuboid intersections or line intersections for hazard detection. TransFuser also introduced the Longest6 benchmark, which consists of longer routes compared to the official CARLA benchmark and is less saturated.

## 3 THE LANGPROP FRAMEWORK

The LangProp framework, visualized in Figure 2, addresses a general task of optimizing code on any given metric of success in a data-driven way, similar to how a neural network is optimized on an objective function. LangProp performs iterative prompting to improve code performance, using the inputs, outputs, exceptions, metric scores, and any environmental feedback to inform the LLM upon updates. The updates in LangProp are performed using a form of an evolutionary algorithm (Bäck & Schwefel, 1993). The following sections describe the key concepts in LangProp in more detail.

### 3.1 MODEL DEFINITION

The LangProp model consists of a setup prompt, an update prompt, and a collection of executable code generated by the LLM, which we refer to as a *policy*. While neural models are parameterized by floating-point weights, the *parameters* of a LangProp model is the collection of policies. Each policy is associated with an executable *script* as well as a statistics tracker, which updates the *priority*, an aggregate measure of the policy's performance with respect to the training objective. The priority is used to rerank the policies so that the best-performing policies are used for updates and inference.

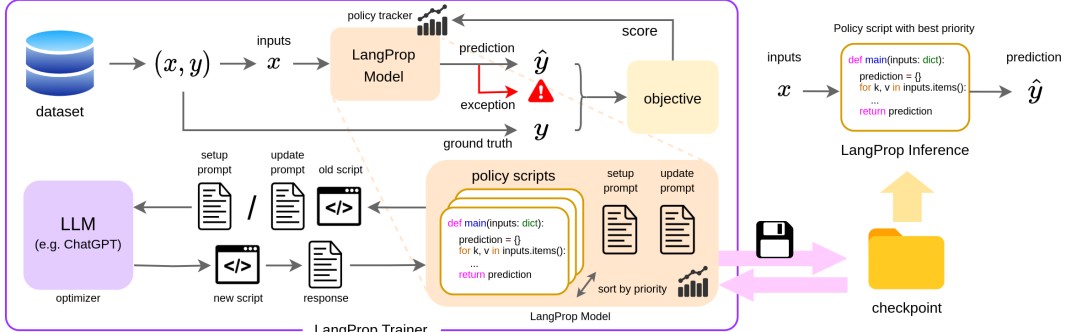

Figure 1: An overview of the LangProp framework, which consists of a LangProp model, an LLM optimizer, and a LangProp trainer. During training, the LLM generates and updates the policy scripts which are evaluated against a training objective. The performances of the policies are monitored and aggregated over time by a policy tracker as *priorities*, which is then used to rerank the policies. Policies with higher priorities are selected for updates, and the best policy is used for inference.

### 3.1.1 POLICY SETUP

The initialization of the policies is done similarly to zero-shot code generation. The definition and specification of the requested function is given, for example, the docstring of the function including the names and types of the inputs and outputs, what the function is supposed to achieve, and a template for the function. We also adopt Chain-of-Thought prompting (Wei et al., 2022). An example of a setup prompt can be found in Appendix A.1. The response from the LLM is parsed to extract the solution code snippet. Multiple responses are collected to ensure the diversity of the initial policies.

### 3.1.2 TRAINING OBJECTIVE

The advantage of LangProp over typical usage of LLMs for code generation is that it performs code optimization in a metric- and data-driven manner. In many tasks, it is easier to provide a dataset of inputs and ground truth corresponding outputs rather than to accurately specify the requirements for a valid solution or write comprehensive unit tests. Similar to how neural networks are trained, the user defines an objective function that measures how accurate the policy prediction is against the ground truth, e.g. L1 or L2 loss. A penalty is given if the policy raises an exception.

### 3.1.3 FORWARD-PASS AND FEEDBACK

Similar to training neural networks, LangProp assumes a dataset of inputs and associated ground truth labels for supervised learning (or rewards/returns for reinforcement learning, discussed in Section 4.3). For every batch update, the inputs are fed into all the policies currently in the LangProp model to make predictions, equivalent to a *forward-pass*. For each policy, the prediction is evaluated by the objective function which returns a *score*. If an exception is raised during execution of a policy script, it is caught by the model and an exception penalty is returned as a score instead.

The execution results, which include the score, exception trace, and any print messages from the execution, are fed back into the model and are recorded by the policy tracker. This is analogous to how parameters in a neural network are assigned gradients during back-propagation (see Appendix A.9). This information stored by the tracker is used in the policy update step in Section 3.1.5.

### 3.1.4 PRIORITY

The priority is, simply put, an average of scores with respect to the training objective. In case a small batch size is required for faster computation, a running average of the scores is used as the priority rather than ranking the policies' performance based on scores from the current batch alone, which may result in highly stochastic results. This is sufficient for supervised learning with a fixed size dataset. As discussed later in Section 4.3, however, a more complex training method such as reinforcement learning or DAgger (Ross et al., 2011) has a non-stationary training distribution.

Therefore, we use exponential averaging with a discount factor of $\gamma \in (0, 1]$ following Equation (1).

$$P_{i,k} = (\sum_{j=1}^{N_k^B} s_{i,j,k} + W_{i,k-1}P_{i,k-1})/(N_k^B + W_{i,k-1}), \quad W_{i,k} = \gamma(N_k^B + W_{i,k-1}) \tag{1}$$

Here, $N_k^B$, $P_{i,k}$ and $W_{i,k}$ are the batch size, priority, and priority weighting of the $k$-th batch for the $i$-th policy, respectively, and $s_{i,k}$ is the objective score of the $i$-th policy for the $j$-th element in the $k$-th batch. Initial conditions are $P_{i,0} = 0$ and $W_{i,0} = 0$. By weighting recent scores higher, we ensure policies with higher priorities have high performance on the most up-to-date dataset.

### 3.1.5 POLICY RERANKING AND UPDATE

This step updates the model based on the most recent forward-backward pass and updated priorities. This corresponds to the optimization step in neural network training, where parameters are updated based on gradients computed on the most recent batch. First, the policies are reranked by the priorities and the top $N^K$ number of policies are kept, out of which the top $N^U$ policies are selected for updates. For each of these policies, the policy tracker storing records of inputs, outputs and scores is queried for the worst-case input-output pairs in the training batch that had the minimum score, along with any exception or print messages during the execution. This information, together with the old policy script, is embedded into the update prompt by a prompt template engine (Section 3.2). The update prompt is passed to the LLM, which returns $N^R$ responses containing new policy scripts.

After the model update, there are $N^U \times N^R$ new policies, as well as up to $N^K$ old policies. To initialize the new policies with sensible priorities, extra forward-backward passes are performed on these policies with the same batch of samples used for the model update. Finally, all policies are sorted according to their priorities, ready for inference or training on a new batch.

### 3.2 PROMPT TEMPLATE ENGINE

During the policy update stage, we require a dynamic prompting mechanism to embed information about the input, predicted output, ground truth, exception, print messages, and the policy script to be revised. The logic to generate these prompts is sometimes complex, for example, predictions are only made when there are no exceptions. To enable flexible prompt generation while avoiding any hardcoding of the prompts in the codebase, we developed a simple yet powerful prompt template that can parse variables, execute Python code embedded within the prompt, and import sub-prompts from other files, and will be included in our open-sourced solution. The update prompt example shown in Appendix A.2 makes extensive use of the policy template engine's capabilities.

### 3.3 TRAINING PARADIGM

LangProp mirrors the code abstraction of PyTorch (Paszke et al., 2019) and PyTorch Lightning (Falcon, 2019) for the module and trainer interfaces, respectively. This allows LangProp to be task-agnostic, making it easily applicable to a range of domains and use cases. Moreover, it helps highlight the similarities between neural network optimization and code optimization using LangProp and facilitates a smooth integration of other training paradigms for neural network training.

Importantly, LangProp's internal implementation does not depend on PyTorch or PyTorch Lightning. LangProp supports PyTorch datasets and data loaders, as well as any iterable dataset object for training and validation. Listing 1 shows an example of a standard LangProp training script.

```
1 train_loader = DataLoader(train_data, batch_size, shuffle=True, collate_fn=lambda x: x)
2 val_loader = DataLoader(val_data, batch_size, shuffle=True, collate_fn=lambda x: x)
3 model = LPModule.from_template(name=model_name, root=model_root)
4 trainer = LPTrainer(model, RunConfig(run_name=run_name))
5 trainer.fit(train_loader, val_loader, epochs=epochs)       # train model
```

Listing 1: Training a LangProp model with a LangProp trainer.

After every training step on a mini-batch, the trainer saves a *checkpoint*, which consists of the setup prompt, update prompt template, the currently kept policy scripts (maximum of $N^K + N^U \times N^R$),

and the statistics monitored by the policy tracker (priorities $P$ and priority weights $W$). Since these can be stored as text or JSON files, the size of a checkpoint is in the order of a few hundred kilobytes. Checkpoints can be used to resume training, fine-tune the model, or for inference.

```
1 model = LPModule.from_checkpoint(checkpoint)          # load checkpoint
2 model.setup(config=RunConfig())
3 prediction = model(*input_args, **input_kwargs)        # make prediction
```

Listing 2: Inference with a pre-trained LangProp model checkpoint.

Listing 2 shows how a LangProp checkpoint can be loaded and used for inference. The policy with the highest priority is used for inference. Since policies are *parameterized* as executable code, the use of an LLM is only required during training, not during inference. Since querying LLMs is both expensive and slow, this is a key advantage of the LangProp approach, which makes integration of LLMs more feasible for real-time applications, such as robotics and autonomous driving.

## 4 LANGPROP APPLIED TO DRIVING IN CARLA

In this section, we describe how the LangProp framework can be used in the context of autonomous driving. We chose the CARLA environment (Dosovitskiy et al., 2017) as a benchmark since (a) autonomous driving requires interpretable driving policies, (b) CARLA has a rich collection of human-implemented expert agents to compare against, and (c) a metric-driven learnable approach would be beneficial since driving decisions such as when to lane-change or to give way are challenging planning problems, and even human-implemented experts have sub-optimal performance.

### 4.1 EXPERT

We implemented our expert agent for data collection and to provide pseudo-ground-truth actions to train the LangProp agent with imitation learning. While TransFuser (Chitta et al., 2022) and TF++ (Jaeger et al., 2023) use a computationally expensive 3D bounding box collision detection algorithm, and InterFuser (Shao et al., 2023) uses line collision which is faster but less accurate, we use an efficient polygon collision detection algorithm between ground-projected bounding boxes. By extrapolating the motion of the ego vehicle and the actors into the future and checking for any polygon intersections, the safety margins to the pedestrians and vehicles are calculated. Together with the distance to the nearest traffic light and/or stop sign, the target speed is determined to give a $2\,s$ margin. Steering is evaluated by calculating the angle to the next waypoint, which is $4\,m$ ahead of the ego vehicle. A PID controller is used for low-level control to convert the target speed and angle to throttle, brake, and steering. For more implementation details, see Appendix B.2.

### 4.2 LANGPROP AGENT

Similarly to our expert and all the baseline experts, we provide privileged information from the CARLA simulator to the agent. While we manually convert the bounding box coordinates of actors in the scene into the ego-relative frame of reference, we let LangProp handle these computations, providing everything in absolute world coordinates. We provide the location, orientation, speed, length, and width of the ego vehicle as well as for other vehicles and pedestrians that are within the range of $50\,m$. Importantly, we do not filter out actors even if they are irrelevant to the driving agent. We also provide the target waypoint ($4\,m$ ahead, used by other baseline experts) and the distances to a red traffic light and stop sign along the current lane if they exist. Given this information, the LangProp policy is expected to return a desired speed level ("MOVE": $6\,m/s$, "SLOW": $1\,m/s$, "STOP": $0\,m/s$)[1] and a turning angle for the ego vehicle. These are passed to an external PID controller to convert them into throttle, brake, and steering. A more detailed explanation of the function definition is given in Listing A.5, which is an extract of the setup prompt used in the LangProp model. Given the function definition as a docstring, an LLM generates policy script

---

[1]While it is straightforward for the policy to directly predict the speed or acceleration as numeric values, this makes the task of designing a suitable loss function for imitation learning more challenging and open-ended. Therefore, we opted for a categorical output which simplifies the scoring function.

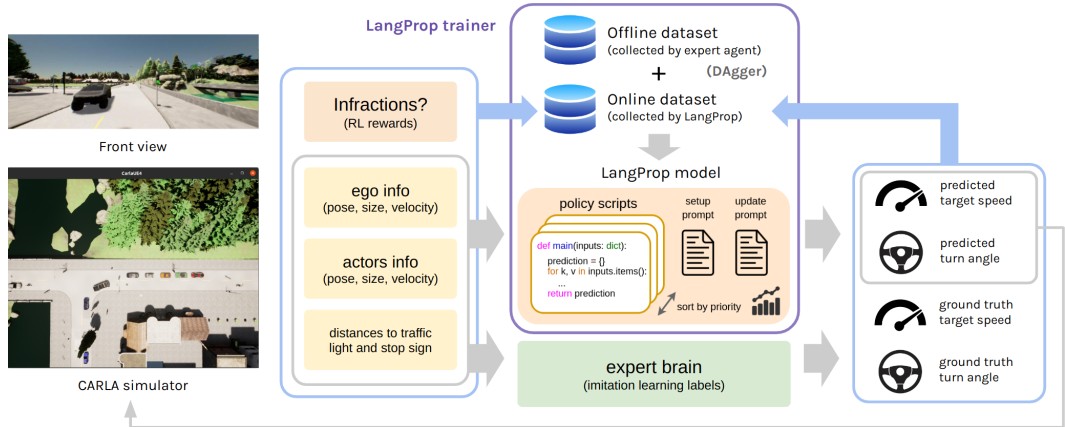

Figure 2: An overview of the LangProp agent training pipeline. The LangProp model is updated on a dataset that includes both offline expert data as well as online LangProp data annotated with expert actions, similar to DAgger. The agent is given negative rewards upon infraction.

candidates that satisfy the specification and updates them following the procedures in Section 3. We use GPT 3.5 Turbo 16k model, provided by OpenAI's Chat Completion API (OpenAI, 2022).

### 4.3 IMITATION LEARNING, DAGGER, AND REINFORCEMENT LEARNING

We explore three major training paradigms often used to train embodied agents - imitation learning (IL), DAgger (Ross et al., 2011), and reinforcement learning (RL). In imitation learning, the accuracy of the policy outputs is measured against ground truth expert actions for a pre-collected dataset. Imitation learning is known to have issues with out-of-distribution inputs at inference time, since the expert's policy is used to collect the training data, while the learned policy is used for rollouts at inference time. DAgger addresses this issue by labeling newly collected *online* data with expert actions, and adding them to the expert-collected *offline* data to form an aggregate replay buffer. Both CARLA and the LangProp agent run at a frame rate of 20 $Hz$. LangProp adds training samples to the replay buffer every 10 frames, and a batch update is performed after every 100 new samples.

While DAgger solves the issue of distribution mismatch, the performance of the learned policy is still upper-bounded by the accuracy of the expert. It also does not take into account that certain inaccuracies are more critical than others. In the context of autonomous driving, actions that result in infractions such as collisions should be heavily penalized. Reinforcement Learning offers a way of training a policy from reward signals from the environment, which is convenient since we can directly assign penalties upon any infractions according to the CARLA leaderboard (CARLA, 2020). While RL typically optimizes for maximum returns (discounted sum of future rewards), we simplify the setting by assigning an infraction penalty if there is an infraction in the next 2 $s$ window. The agent monitors infractions every 10 frames, and triggers an update upon infractions.

Since infraction penalties are very sparse, and will become rarer as the policies improve, we adopt two strategies; (a) we combine RL training with imitation learning training that provides denser signals, and (b) we sample training data with infractions with 100 times higher sampling probability. The expert is only imitated upon no infractions, or if the expert was not the behavior policy which incurred the infraction, and an infraction cost is only given when the current policy takes the same action as the behavioral policy which caused the infraction when the expert chose a different action. For more details on the training objective, see Appendix C.2.

## 5 EXPERIMENTS

We compared our LangProp agent against RL experts with privileged information (Roach (Zhang et al., 2021), TCP (Wu et al., 2022)) as well as human-implemented experts (TransFuser (Chitta et al., 2022), InterFuser (Shao et al., 2023), TF++ (Jaeger et al., 2023), ours). We used the official training and testing routes provided by the CARLA leaderboard (CARLA, 2020), as well as

Table 1: Driving performance of expert drivers in CARLA version 0.9.10. The driving score is a product of the route completion percentage $\bar{R}$ and the infraction factor $\bar{I}$. IL and RL stand for imitation learning and reinforcement learning. DAgger uses both online and offline data.

| Method | Training routes | | | Testing routes | | | Longest6 | | |
|---|---|---|---|---|---|---|---|---|---|
| | Score $\uparrow$ | $\bar{R}\uparrow$ | $\bar{I}\uparrow$ | Score $\uparrow$ | $\bar{R}\uparrow$ | $\bar{I}\uparrow$ | Score $\uparrow$ | $\bar{R}\uparrow$ | $\bar{I}\uparrow$ |
| Roach expert | 57.8 | 95.9 | 0.61 | 63.4 | 98.8 | 0.64 | 54.9 | 81.7 | 0.67 |
| TCP expert | 64.3 | 92.3 | 0.71 | 72.9 | 93.2 | 0.77 | 46.9 | 63.1 | 0.76 |
| TransFuser expert | 69.8 | 94.5 | 0.74 | 73.1 | 91.3 | 0.80 | 70.8 | 81.2 | 0.88 |
| InterFuser expert | 69.6 | 83.1 | 0.86 | 78.6 | 81.7 | 0.97 | 48.0 | 56.0 | 0.89 |
| TF++ expert | **90.8** | 95.9 | 0.94 | 86.1 | 91.5 | 0.94 | **76.4** | 84.4 | 0.90 |
| **Our expert** | 88.9 | 92.8 | 0.95 | **95.2** | 98.3 | 0.97 | 72.7 | 78.6 | 0.92 |
| LangProp: Offline IL | 0.07 | 0.37 | 0.97 | 0.00 | 0.00 | 1.00 | 0.00 | 0.00 | 1.00 |
| LangProp: DAgger IL | 36.2 | 94.5 | 0.40 | 41.3 | 95.3 | 0.44 | 22.6 | 87.4 | 0.30 |
| LangProp: DAgger IL/RL | 64.2 | 90.0 | 0.72 | 61.2 | 95.2 | 0.64 | 43.7 | 71.1 | 0.65 |
| LangProp: Online IL/RL | **70.3** | 90.5 | 0.78 | **80.9** | 92.0 | 0.89 | **55.0** | 75.7 | 0.73 |

the Longest6 benchmark (Chitta et al., 2022) that has longer routes with denser traffic. See Appendix D.1 for more details on the benchmark and the routes and towns used. For the LangProp agent, only the training routes are used for imitation/reinforcement learning at training time, and the saved checkpoints are used for inference during evaluation runs. The results are shown in Table 1.

## 5.1 EXPERT AND LANGPROP AGENTS

Our expert and the TF++ expert significantly outperformed all other expert agents in all routes, and our expert outperformed TF++ by a margin on the test routes. The core collision avoidance logic is just 100 lines of code, with additional preprocessing and tooling for data collection. From the breakdown of the scores, our expert seems to prioritize safer driving with fewer infractions (higher infraction factor $\bar{I}$) by trading off route completion compared to TF++ in the Longest6 benchmark.

For the LangProp agent, we observe that training using offline samples, DAgger, and online samples improves performance in this order. Adding the infraction penalties as an additional reinforcement learning objective further improved the performance. The best-performing agent, LangProp trained on online data with IL and RL, achieved better performance than the Roach expert (trained with PPO) as well as the TransFuser and InterFuser experts (both written by researchers) on all benchmarks apart from TransFuser on the Longest6 benchmark.

The result has two important implications. Firstly, the code selection metric (the training objective) plays a large role in the ultimate performance of the code. This is an important finding since prior work on code generation mostly focused on error correction given exceptions. Our results demonstrate that for complex tasks, it is important to treat code generation as an iterative optimization process rather than a zero-shot task. Secondly, training using LangProp exhibits similar characteristics as training in deep learning; in deep learning, it is a well-studied problem that policies trained with imitation learning on offline datasets do not generalize to out-of-distribution online data. DAgger and reinforcement learning are two of the common ways of addressing this problem. Our results show that these training paradigms can also be effective when used in LangProp.

## 5.2 DEMONSTRATION OF CAUSAL CONFUSION WHEN TRAINED OFFLINE

A common failure mode of offline trained models was that the agent remained stationary indefinitely until the timeout was reached. Upon inspection of the policy code that was generated, we were able to identify the failure to be a phenomenon known as causal confusion in imitation learning (De Haan et al., 2019). A snippet of code responsible for such failure in one of the runs is shown in Listing 3.

This exemplifies the interpretability of LangProp models, allowing us to directly assess the source of failure. The code predicts 0 speed when the agent's current speed is already close to 0. Note that this is not a failure of the LangProp algorithm, but due to such a policy maximizing the imitation

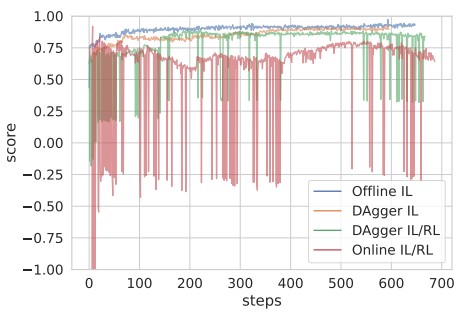

(a) training scores on the replay buffer

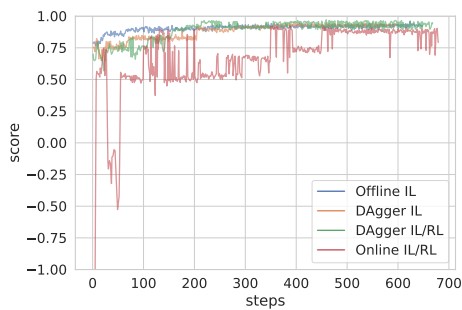

(b) validation scores on the offline dataset

Figure 3: Training curves for the different training methods of the LangProp agent. The training scores are evaluated on 1000 samples from the offline training dataset and/or online replay buffer, and the validation scores are evaluated on 1000 samples from the offline validation dataset. Updates are performed every 1000 frames of agent driving, as well as upon infractions in the RL setting. The score is in the range of $[-10, 1]$ due to exception penalties. We limit the axis to $[-1, 1]$ in the plots.

learning objective on an offline dataset, bypassing the need to learn a more complex policy. This phenomenon is commonly researched in the context of *deep* imitation learning, and can be avoided by employing training on online data, e.g. using DAgger or RL. We believe our work to be the first to report a similar phenomenon using LLMs for policy optimization.

```
1  # General rule: if the ego vehicle is stopped or moving very slowly, set the speed
↪     level to "STOP"
2  if np.abs(scene_info["ego_forward_speed"]) < DELTA_V_THRESHOLD:
3      speed_level = "STOP"
```

Listing 3: Identifying causal confusion in the policy when trained purely offline

## 5.3 ANALYSIS OF TRAINING METHODS

The use of online training samples alleviated the issue of causal confusion, leading to selecting policies where the agent has a sensible driving performance. This is because if the agent remains stationary, those samples will accumulate in the replay buffer, resulting in a lower priority for the causally confused policy. Comparing the results in Table 1 and the validation scores in Figure 3b, it seems that the scores on the offline dataset are not indicative of the agent's driving performance. From the training scores on the replay buffer and/or offline dataset in Figure 3a, we see that the agents trained with RL on infractions have spikes corresponding to infractions. This is due to over-sampling infractions when they occur, allowing the policy update to immediately address the issue. DAgger has a milder response compared to training just on online data because the offline dataset does not include on-policy infractions. The higher rate of infractions in the training distribution may be why the online trained agent has a lower training score but has a higher driving performance.

## 6 CONCLUSION

We presented LangProp, a framework that uses LLMs for data-driven code optimization, and demonstrated its capability of generating driving policies in CARLA. We showed that classical training paradigms such as imitation learning, DAgger, and reinforcement learning directly translate to training with LangProp, and the choices of the objective function and the training data distribution can be used to guide which policies are selected. Since numerous candidate solutions satisfy the code specification, automatically optimizing the code to maximize a given performance metric has been a key missing feature in few-shot code generation. The LangProp framework provides this feature by reformulating the machine learning training paradigm in the context of using LLMs as code optimizers and treating policy code as parameters of the model. We believe that the LangProp paradigm opens up many possibilities for data-driven machine learning with more interpretability and transparency.

## REPRODUCIBILITY STATEMENT

We will open-source the code both for the general LangProp framework, as well as the code for training and evaluating the LangProp agent in CARLA. More details of the implementation and design decisions can be found in the appendices.

For the ICLR 2024 conference submission, supplementary materials can be found at `https://github.com/langprop-iclr24/LangProp/`, which includes the code, pre-trained checkpoints using LangProp, videos of sample runs by the LangProp agent. We also include self-contained minimal examples of applying LangProp to tasks such as Sudoku and CartPole.

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

# Appendices

## A    LANGPROP MODEL AND PROMPT DEFINITIONS

LangProp as a framework can be used to optimize a diverse range of code optimization problems. The functionality of the model is determined by the choices in the setup prompt, the update prompt, and the dataset that the LangProp model is trained on.

### A.1    POLICY SETUP PROMPT EXAMPLE

We provide a simplified example of learning a function to compute the factorial for a given number to show the generality of the framework. The setup prompt should include the specification of the function's inputs and outputs and their types in the form of a docstring.

```
1  I am developing code to evaluate the factorial of an integer input.
2  Here is the definition of the function.
3
4  ```
5  Given a non-negative integer, return the factorial of that integer.
6
7  Args:
8      - number: int       # Has to be non-negative
9
10 Returns:
11     - factorial: int    # Factorial of input number
12 ```
13
14 This is a template of the code.
15
16 ```python
17 def {{ function_name }}(number: int) -> int:
18     # Write code here
19     return output
20 ```
21
22 Please do the following:
23 Step 1. Describe step by step what the code should do in order to achieve its task.
24 Step 2. Provide a python code solution that implements your strategy, including all
   ↪  necessary import statements.
```

Listing A.1: Setup prompt template for a simple factorial calculator

## A.2 POLICY UPDATE PROMPT EXAMPLE

The prompt used to update the policy contains the same information as the setup prompt, but in addition, has example inputs and outputs where the code had failed to produce a valid prediction. If there was an exception or printed messages during the execution of the code, this will also be provided as feedback. The LLM is asked to identify the source of the sub-optimal performance and rewrite the code to achieve a higher score.

```
1  I am developing code to evaluate the factorial of an integer input.
2  Here is the definition of the function.
3
4  ```
5  Given a non-negative integer, return the factorial of that integer.
6
7  Args:
8      - number: int       # Has to be non-negative
9
10 Returns:
11     - factorial: int    # Factorial of input number
12 ```
13
14 Here is an example code that I have written. However, it is not working as expected.
15
16 ```python
17 {{ code }}
18 ```
19
20 I executed the code, and got an accuracy of {{ int(avg_score * 100) }}%.
21
22 $begin
23 if printed:
24     print("There was a print message saying: {{ printed }}")
25 if exception:
26     print("""The code failed to run because there was an exception. The exception
    ↪   message was as follows: {{ exception }}""")
27     print("Resolving this exception is the top priority.")
28 else:
29     print("""
30 The code produced incorrect results for the following inputs. The prediction, ground
    ↪   truth label and score were as follows.
31
32 Inputs: {{ args[0] }}
33 Incorrect prediction: factorial = {{ outputs }}""")
34     if {{ label }}:
35         print("Ground truth label: factorial = {{ label }}")
36     print("Score: {{ int(score * 100) }}%")
37 $end
38
39 $begin
40 if feedback:
41     print("""{{ feedback }}""")
42 $end
43
44 Please do the following:
45
46 $begin
47 if exception:
48     print("Step 1. Look at the error message carefully and identify the reason why the
    ↪   code failed, and how it can be corrected.")
49 else:
50     print("Step 1. Given the example input and output, identify the reason why the code
    ↪   made a wrong prediction, and how it can be corrected to achieve a good driving
    ↪   score.")
51 $end
52
53 Step 2. Describe step by step what the code should do in order to achieve its task.
54 Step 3. Please rewrite the python function `{{ function_name }}` to achieve a higher
    ↪   score, including all necessary import statements.
```

Listing A.2: Update prompt template for a simple factorial calculator

### A.3 MODEL FORWARD PASS DEFINITION

The LangProp module captures printed outputs and exceptions and stores them in the policy tracker along with the corresponding inputs during a forward pass. The Python code snippet extracted from the LLM's response and saved as a text string is executed using the `exec` function in Python. The local scope variables can be accessed via `locals`.

```python
class LPModule:
    ...

    def __call__(self, *args, **kwargs) -> Any:
        if not self.training:
            return self.forward(self.script_records[0].script, *args, **kwargs)

        inputs = (args, kwargs)
        script = self.run_config.active_tracker.record.script
        with CapturePrint() as p:
            try:
                output = self.forward(script, *args, **kwargs)
                self.run_config.active_tracker.forward(inputs, output, "\n".join(p))
            except KeyboardInterrupt as e:
                raise e
            except Exception as e:
                trace = "\n".join(traceback.format_exc().split('\n')[-3:])
                detail = f"""{type(e).__name__}: {trace}"""
                self.run_config.active_tracker.store_exception(inputs, e, detail,
                    "\n".join(p))
                raise e
        return output

    def forward(self, script, *args, **kwargs):
        exec(script, locals(), locals())
        output = locals()[self.name](*deepcopy(args), **deepcopy(kwargs))
        return output
```

Listing A.3: Forward passing mechanism of the LangProp module (extract)

### A.4 TRAINER FORWARD-BACKWARD DEFINITION

The trainer has a similar abstraction to deep learning training. At every step, it triggers a forward method that calls the policy and stores the inputs, the policy's prediction, and the expected output, and a backward method that updates the policy tracker with the scores, exceptions, or any feedback.

```python
class LPTrainer:
    ...

    def step(self, tracker: RecordTracker, func_args, func_kwargs, label, feedback=""):
        with self.run_config.activate(tracker):
            score, exception_detail = self.forward(func_args, func_kwargs, label)
            tracker.backward(score, label, feedback + exception_detail)

    def forward(self, func_args, func_kwargs, label):
        try:
            with set_timeout(self.run_config.forward_timeout):
                output = self.module(*func_args, **func_kwargs)
            self.test_output(output, func_args, func_kwargs, label)
            score = self.score(output, label)
            exception_detail = ""
        except KeyboardInterrupt as e:
            raise e
        except Exception as e:
            score = self.run_config.exception_score
            trace = "\n".join(traceback.format_exc().split('\n')[-3:])
            exception_detail = f"""\nThere was an exception of the
                following:\n{type(e).__name__}: {trace}"""
        return score, exception_detail
```

Listing A.4: Forward-backward pass in the LangProp Trainer (extract)

## A.5  POLICY DEFINITION FOR THE LANGPROP DRIVING AGENT IN CARLA

The driving policy is given the location, orientation, speed, length, and width of the ego vehicle, other vehicles and pedestrians in the scene, the distances to the next red traffic light and stop sign, and the target waypoint ($4\ m$ ahead, used by other baseline experts), all in absolute world coordinates.

```
1   ```
2   Args:
3       - scene_info: dict
4           Contains the following information:
5           {
6               "ego_location_world_coord": np.ndarray,        # numpy array of shape (2,)
                ↪  which contains (x, y) of the center location of the ego vehicle in
                ↪  world coordinates given in [m]
7               "ego_target_location_world_coord": np.ndarray,  # numpy array of shape (2,)
                ↪  which contains (x, y) of the target location of the ego vehicle in
                ↪  world coordinates given in [m]
8               "ego_orientation_unit_vector": np.ndarray,      # numpy array of shape (2,)
                ↪  which contains (x, y) of unit vector orientation of the ego vehicle in
                ↪  world coordinates. The vehicle moves in the direction of the
                ↪  orientation.
9               "ego_forward_speed": float,                     # the speed of the ego
                ↪  vehicle given in [m/s].
10              "ego_length": float,                            # length of the ego vehicle
                ↪  in the orientation direction, given in [m/s].
11              "ego_width": float,                             # width of the ego vehicle
                ↪  perpendicular to the orientation direction, given in [m].
12              "distance_to_red_light": Union[float, None],    # distance to red light
                ↪  given in [m]. None if no traffic lights are affecting the ego vehicle
13              "distance_to_stop_sign": Union[float, None],    # distance to stop sign
                ↪  given in [m]. None if no stop signs are affecting the ego vehicle
14              "vehicles": {                   # dictionary of nearby vehicles
15                  <vehicle_id: int>:  {
16                      "location_world_coord": np.ndarray,     # numpy array of shape (2,)
                        ↪  which contains (x, y) of the center location of vehicle
                        ↪  <vehicle_id> in world coordinates given in [m]
17                      "orientation_unit_vector": np.ndarray,  # numpy array of shape (2,)
                        ↪  which contains (x, y) of unit vector orientation of vehicle
                        ↪  <vehicle_id> in world coordinates. The vehicle moves in the
                        ↪  direction of the orientation.
18                      "forward_speed": float,                 # speed of vehicle
                        ↪  <vehicle_id> given in [m/s].
19                      "forward_length": float,                # length of the vehicle
                        ↪  <vehicle_id> along the orientation direction, given in [m].
20                      "sideways_width": float,                # width of the vehicle
                        ↪  <vehicle_id> perpendicular to the orientation direction, given
                        ↪  in [m].
21                  },
22              },
23              "pedestrians": {                # dictionary of nearby pedestrians
24                  <pedestrian_id: int>:  {
25                      "location_world_coord": np.ndarray,     # numpy array of shape (2,)
                        ↪  which contains (x, y) of the center location of pedestrian
                        ↪  <pedestrian_id> in world coordinates given in [m]
26                      "orientation_unit_vector": np.ndarray,  # numpy array of shape (2,)
                        ↪  which contains (x, y) of unit vector orientation of pedestrian
                        ↪  <pedestrian_id> in world coordinates. The vehicle moves in the
                        ↪  direction of the orientation.
27                      "forward_speed": float,                 # speed of pedestrian
                        ↪  <pedestrian_id> relative to the orientation given in [m/s].
28                      "forward_length": float,                # length of the pedestrian
                        ↪  <pedestrian_id> along the orientation direction, given in [m].
29                      "sideways_width": float,                # width of the pedestrian
                        ↪  <pedestrian_id> perpendicular to the orientation direction,
                        ↪  given in [m].
30                  },
31              }
32          }
33
34  Returns:
35      - speed_level: str        # Choose from ("MOVE", "SLOW", "STOP").
36      - turn_angle: float       # Predicted turn angle of the ego vehicle to reach the
        ↪  target waypoint in [degrees]. The range should be between -180 to 180 degrees
37  ```
```

Listing A.5: Docstring given as part of the setup prompt for the LangProp agent

A.6 Notes on specifying the policy

One of the challenges in the early stages of the project was in specifying the inputs and outputs of the function. Most of the failures in learning a policy were due to misspecification of the inputs, rather than a fundamental problem with the LLM or with LangProp. For instance, we found that it is crucial to specify the units of the input values, e.g. $m/s$, which allowed the LLM to choose sensible values for some internal parameters. It was also important to name input variables explicitly such that it is clear whether the coordinates are given as absolute world coordinates or coordinates relative to the ego vehicle. A useful property of LangProp is that because the LLM has some understanding of the world from natural language, it can easily incorporate this knowledge when generating the code, constraining the search space of feasible code. We can further guide the LLM to generate policies with certain characteristics, e.g. having a larger safety margin, by expressing our preferences in the prompts. This adds to the benefits of the LangProp approach, where it is easier to encourage policies to exhibit certain behaviors.

A.7 Details of the prompt template engine

In the template engine, every line that begins with "#" is treated as comments. Every line that begins with "$ " or line blocks in between "$begin" and "$end" are treated as executable Python code, as well as everything surrounded by {{ }} in a single line. If a "print" function is used within the prompt template, it will execute the Python code inside the print function and render the resulting string as a part of the prompt. Variables can be passed to the prompt template engine, and are made accessible in the local scope of the prompt template.

As an example, consider the following prompt template.

```
1 {{" and ".join(p for p in people)}} {{"are" if len(people) > 1 else "is"}} work here.
2 $begin
3 for i, p in enumerate(people):
4     print(f"{p} is employee number {i + 1}.")
5 $end
```

Listing A.6: Example template

If the prompt template engine is called with the arguments `read_template("example", people=["Tom", "Jerry"])`, this resolves to: "Tom and Jerry work here.\nTom is employee number 1.\nJerry is employee number 2.".

A.8 How to choose the priority discount factor

How the priorities of the policies are calculated has a large effect on the final performance of the trained LangProp model. For a stationary training distribution (e.g. supervised learning on a fixed offline dataset), whether one uses the immediate average score, a running average, or an exponential average does not make a difference except that just using the immediate average score results in a more stochastic result due to fewer numbers of samples. If the computational resources and time are not constrained, one could increase the batch size and just use the immediate average score. If these are constrained, one may adopt a running average with smaller batch sizes. This works when the training distribution is stationary and there are no other changing components other than the policy currently training.

If the training distribution changes or the policy consists of multiple chained modules, each with a learnable sub-policy, we can no longer use a simple running average but have to use either the scores evaluated on a single large batch or the exponential averaging scheme. The current implementation of LangProp does not support multiple chained modules, but is a foreseeable and natural extension to the framework. Changes in the training distribution are expected in DAgger or reinforcement learning. For training our LangProp agent in Section 4.3, we used a discount factor $\gamma = 0$, effectively only using the immediate average scores evaluated on a freshly sampled batch. This is because forward passes through the LangProp driving policies are fast due to not having any complex components so we could afford to have a large batch size. However, in applications where

forward passes are expensive and the batch size must be small, using exponential averaging with a non-zero discount factor $\gamma$ is recommended.

### A.9 USE OF THE TERM "BACK-PROPAGATION"

The current LangProp implementation is limited to an update of a single module, i.e. it does not yet accommodate for chaining of modules. We have explored this path by making the LLM generate docstrings of helper functions so that submodules can be instantiated, and track priorities also for submodules. However, version tracking of submodules and the mechanism of providing feedback for submodule updates were substantial challenges. LangProp v1 does not implement the full back-propagation algorithm, but we refer to a single-layer feedback operation as *back-prop* to highlight the similarities and encourage future research in this area.

### A.10 SELF-CONTAINED MINIMAL EXAMPLES OF APPLYING LANGPROP

We include self-contained examples of applying LangProp to (a) Sudoku and (b) CartPole.

For Sudoku, we solve a generalized sudoku puzzle that consists of $W \times H$ subblocks, each with $H \times W$ elements, where $H$ and $W$ represent height and width, respectively. Due to the complexity of the task specification, we found that the LLM queried zero-shot occasionally failed on the first attempt, confusing the task with a standard $3 \times 3$ sudoku, but using LangProp allowed us to filter out incorrect results and arrive at a fully working solution.

For CartPole, we provided the observation and action specifications in the Gymnasium documentation for CartPole-v1. Queried zero-shot, the LLM generated a solution which is simplistic and does not balance the CartPole, achieving a score of 9.9 out of 500. With a simple Monte-Carlo method of optimizing the policy for the total rewards, we obtained improved policies using LangProp, achieving the maximum score of 500.0 (evaluated over 100 runs).

Implementations, prompts, checkpoints, and comparisons of zero-shot and trained policies are available in the open-sourced repository.

## B   DATA COLLECTION

### B.1   DATA AGENT

To standardize the data collection and evaluation pipeline for both our expert agent and our Lang-Prop agent, we implement a generic `DataAgent` that collects basic information of the CARLA environment which can be used. These are the 3D bounding box coordinates of the actors in the scene (pedestrians, vehicles, traffic lights, and stop signs), the velocity of the pedestrians and vehicles, distances to the next traffic light and stop sign in the current lane, and the next waypoint to navigate towards. In addition, it also collects the RGB, depth, lidar, segmentation, top-down view, and the expert's control actions which can be used to train image-based driving policies. We created this standardized data collection agent which is decoupled from our expert agent and the LangProp agent, and has the option of turning off sensors that are not used for data collection to save computation time and data storage.

The data collection agent itself does not have a driving policy. It expects a separate `AgentBrain` that takes a dictionary of scene information curated by the data agent as input and outputs a vehicle control action (throttle, brake, and steering). All driving agents inherit from the `DataAgent` class, each with an `AgentBrain` that implements its driving policy. It is also possible to chain multiple agent brains as an array, where the previous agent brain's control decision is provided as an additional input to the next agent brain. This is useful for our DAgger and online agents, which require expert supervision during online rollouts.

### B.2   EXPERT AGENT

Our expert agent only uses the data collected by the data agent to ensure that the LangProp agent has access to the same privileged information as the expert agent. For every interval of $0.25\ s$ up to $2\ s$ into the future, we evaluate whether the ego vehicle polygon will intersect any of the actor

polygons, assuming that the ego vehicle will maintain velocity, and the other actors will move in the current orientation with a speed less than or equal to the current speed. The ego vehicle polygon is padded forward by $2\,m$, and by $2\,m$ either left or right upon lane changes. Apart from lane changing, only actors that are ahead of the ego vehicle are considered, i.e. with a field of view of $180°$. The traffic light and stop sign that affect the vehicle are identified by querying the associated waypoints in the CARLA simulator. For pedestrians, vehicles, traffic light, and stop sign, the distances to the obstacles are calculated. The normal driving speed is $6\,m/s$ ("MOVE"). If any of the distances are reachable within $2\,s$ with a $2\,m$ margin ("SLOW"), the target speed is set to the speed which allows a $2\,s$ margin, and if the distance is below $2\,m$ ("STOP"), the target speed is set to $0\,m/s$. Steering is evaluated by calculating the angle to the next waypoint, which is $4\,m$ ahead of the current position of the ego vehicle. A PID controller is used for low-level control to convert the target speed and angle to throttle, brake, and steering.

## C  TRAINING THE LANGPROP AGENT

### C.1  TRAINING STRATEGY

For all the LangProp agents, the training data is collected only on the training routes in CARLA leaderboard (CARLA, 2020), and data collected on the test routes by the expert agent with expert action labels is used as the validation dataset. See Appendix D.1 for more details on the routes. For the LangProp agent trained offline, we only use data collected by the expert agent as training data. For the online training, we only use data collected by the current LangProp model's inference policy, i.e. the policy code with the highest priority at the time of rollout. For DAgger training, we have a split of 1000 training samples collected offline and 1000 samples collected online in every replay batch to evaluate the objective score. Strictly speaking, DAgger (Ross et al., 2011) should incrementally add new online samples to a buffer initialized with offline samples. However, we found that this prevents the LangProp model from learning from infractions during the early stages of the training, since online samples with infractions are the minority of all the samples. For this reason, we maintained an even split between offline and online samples throughout the training, with a sampling weight of 100 for samples with infractions. Sampling is without replacements, so that a particular training sample is only sampled once per replay batch.

### C.2  TRAINING OBJECTIVE

The training objective for the LangProp driving agent is given as Equation (2),

$$
\begin{aligned}
S(a^\pi, a^{\pi_e}, a^{\pi_b}, I, E) = {}& \mathbb{1}\big[(a^\pi_{\text{speed}} = a^{\pi_e}_{\text{speed}}) \wedge [\neg I \vee \{(a^\pi_{\text{speed}} \neq a^{\pi_b}_{\text{speed}}) \wedge (a^{\pi_e}_{\text{speed}} \neq a^{\pi_b}_{\text{speed}})\}]\big] \\
& + r_{\text{infrac}} \mathbb{1}(I \wedge (a^\pi_{\text{speed}} = a^{\pi_b}_{\text{speed}}) \wedge (a^{\pi_e}_{\text{speed}} \neq a^{\pi_b}_{\text{speed}})) \\
& + r_{\text{angle}} \mathbb{1}(|a^\pi_{\text{angle}} - a^{\pi_e}_{\text{angle}}| > \theta_{\text{max}}) + r_{\text{error}} \mathbb{1}(E)
\end{aligned}
\tag{2}
$$

where $a^\pi$, $a^{\pi_e}$ and $a^{\pi_b}$ are actions taken by the current policy, expert policy, and behavior policy used to collect the training sample, respectively, $I$ and $E$ are boolean variables for infraction and exception occurrences, $r_{\text{infrac}} = r_{\text{error}} = r_{\text{angle}} = -10$ are penalties for infraction, exception, and exceeding angle error of $\theta_{\text{max}} = 10°$, and $\mathbb{1}$ equates to 1 if the boolean argument is true, and 0 otherwise. The expert is only imitated when there are no infractions, or if the expert was not the behavior policy that incurred the infraction, and an infraction cost is only given when the current policy takes the same action as the behavioral policy that caused the infraction when the expert chose a different action.

### C.3  HYPERPARAMETERS

Notable training hyperparameters are the number of policies chosen for updates $N^U = 2$, the number of responses per query $N^R = 2$, the number of policies to keep $N^K = 20$, the frequency of batch updates (every 100 new samples in the replay buffer), batch sizes for online replay data (1000) and offline expert data (1000), the sampling weight for infractions (100), and the infraction, exception, and angle penalties ($r_{\text{infrac}} = r_{\text{error}} = r_{\text{angle}} = -10$). For better performance, it is possible to

Table 2: A breakdown of the number of routes per town, the average length of the routes per town, and traffic density for the training routes, testing routes, and the Longest6 benchmark.

| Routes | Training routes | | | Testing routes | | | Longest6 | | |
|---|---|---|---|---|---|---|---|---|---|
| | count | avg. dist. | density | count | avg. dist. | density | count | avg. dist. | density |
| Town 1 | 10 | 776.3 | 120 | - | - | 120 | 6 | 898.8 | 500 |
| Town 2 | - | - | 100 | 6 | 911.7 | 100 | 6 | 911.7 | 500 |
| Town 3 | 20 | 1392.5 | 120 | - | - | 120 | 6 | 1797.5 | 500 |
| Town 4 | 10 | 2262.6 | 200 | 10 | 2177.8 | 200 | 6 | 2102.4 | 500 |
| Town 5 | - | - | 120 | 10 | 1230.1 | 120 | 6 | 1444.7 | 500 |
| Town 6 | 10 | 1915.4 | 150 | - | - | 150 | 6 | 2116.7 | 500 |

increase $N^U$, $N^R$, and $N^K$, but with a trade-off of computational time and the cost of using OpenAI API. With our experiment setting, around 700 training steps are taken, 1400 queries are made, and 2800 responses are received from GPT 3.5 per training job, which costs roughly \$150.

## D  EVALUATION

### D.1  CARLA BENCHMARK, ROUTES AND TOWNS

The driving scores are computed by the CARLA leaderboard evaluator (CARLA, 2020), using the official training and test routes, and the Longest6 benchmark provided by Chitta et al. (2022). There are towns $1-6$ across the benchmarks. Towns $7-10$ are also used in the official online leaderboard. A breakdown of routes for each benchmark is shown in Table 2. Towns 2 and 5 are withheld in the training routes and only appear in the testing routes and the Longest6 benchmark. The Longest6 benchmark has longer routes with denser traffic.

The main metric of the leaderboard is the driving score, which is evaluated as $\frac{1}{N}\sum_i^N (R_i I_i)$, where $i$ denotes the index of the $N$ routes used for evaluation, $R_i$ is the percentage of route completion of the $i$-th route, and $I_i$ is the infraction factor of the $i$-th route. The infraction factor is a product of infraction coefficients for pedestrian collision (0.5), vehicle collision (0.60), collision with static objects (0.65), running a red light (0.70), and running a stop sign (0.80). The driving score per route is equal to the route completion $R_i$ when there are no infractions, and is discounted for every infraction by a corresponding infraction factor. Note that in the Longest6 benchmark, the authors decided to remove the stop sign penalty by setting its infraction coefficient to 1.0, which we adhere to in our experiments.

### D.2  SOFTWARE DETAILS

We use CARLA version 0.9.10 for the experiments to maintain consistency with other baseline experts that assume this version. Our expert has been tested both on CARLA version 0.9.10 and version 0.9.11. For LangProp training, we used GPT 3.5 Turbo 16k chat completion API by OpenAI. We used the 16k-token model since the update prompt often exceeds the 8k-token context size of a smaller model.

## E  DRIVING CODE GENERATED BY LANGPROP

We show an example driving policy generated using LangProp, trained with both imitation learning and reinforcement learning, as described in Section 4.3. Please refer to our open-sourced code repository for the full prompts and code used to train the policy, and pre-trained checkpoints for each training setting used in our evaluation.

```python
import numpy as np
from typing import List, Tuple, Union, Dict

def predict_speed_and_steering(scene_info: Dict, stopping_time_margin=2.0,
```

```
5                                    reaction_time=1.0, stopping_speed=0.1, slow_speed=2.5,
                                 ↪   safety_margin=2.0) -> Tuple[str, float]:
6        """
7        Given the state of the scene, find the required speed level and turn angle of the ego
     ↪   vehicle.
8
9        Args:
10           scene_info (dict): Contains information about the scene.
11           stopping_time_margin (float): Time margin to determine when to stop in seconds
     ↪   [default: 2.0]
12           reaction_time (float): Time taken to react to external events in seconds [default:
     ↪   1.0]
13           stopping_speed (float): Minimum speed below which the vehicle should stop in m/s
     ↪   [default: 0.1]
14           slow_speed (float): Speed at which the vehicle should slow down in m/s [default:
     ↪   2.5]
15           safety_margin (float): Safety margin in meters between the vehicle and other
     ↪   objects [default: 2.0]
16
17       Returns:
18           (str, float): A tuple containing the required speed level and turn angle of the ego
     ↪   vehicle.
19                         The speed level can be "MOVE", "SLOW", or "STOP". The turn angle
     ↪   should be a float between -180 to 180 degrees.
20       """
21       # Step 1: Given the input dictionary, extract all the relevant parameters.
22       ego_location = np.array(scene_info['ego_location_world_coord'])
23       ego_target_location = np.array(scene_info['ego_target_location_world_coord'])
24       ego_orientation = np.array(scene_info['ego_orientation_unit_vector'])
25       ego_forward_speed = scene_info['ego_forward_speed']
26       ego_length = scene_info['ego_length'] + safety_margin
27       ego_width = scene_info['ego_width'] + safety_margin
28       vehicles = scene_info['vehicles']
29       pedestrians = scene_info['pedestrians']
30       distance_to_stop_sign = scene_info['distance_to_stop_sign']
31       distance_to_red_light = scene_info['distance_to_red_light']
32
33       # Step 2: Convert all coordinates for vehicles and pedestrians to relative coordinates
     ↪   in terms of the ego vehicle orientation.
34       vehicles_rel = {}
35       for k, v in vehicles.items():
36           location = np.array(v['location_world_coord']) - ego_location
37           orientation = np.array(v['orientation_unit_vector'])
38           speed = v['forward_speed']
39           length = v['forward_length'] + v['sideways_width'] + safety_margin
40           width = v['sideways_width'] + safety_margin
41           location_rel = np.dot(location, ego_orientation),
     ↪   np.abs(np.dot([-ego_orientation[1], ego_orientation[0]], location))
42           if location_rel[0] > 0:
43               vehicles_rel[k] = {'location_rel': location_rel, 'speed': speed, 'length':
     ↪   length, 'width': width}
44
45       pedestrians_rel = {}
46       for k, v in pedestrians.items():
47           location = np.array(v['location_world_coord']) - ego_location
48           orientation = np.array(v['orientation_unit_vector'])
49           speed = v['forward_speed']
50           length = v['forward_length'] + safety_margin
51           width = v['sideways_width'] + safety_margin
52           location_rel = np.dot(location, ego_orientation),
     ↪   np.abs(np.dot([-ego_orientation[1], ego_orientation[0]], location))
53           if location_rel[0] > 0:
54               pedestrians_rel[k] = {'location_rel': location_rel, 'speed': speed, 'length':
     ↪   length, 'width': width}
55
56       # Step 3: Compute the Euclidean distance from the ego location to the target location.
57       distance_to_target = np.linalg.norm(ego_target_location - ego_location)
58
59       # Step 4: Calculate the threshold stopping distance and threshold slow distance based
     ↪   on the current speed.
60       if ego_forward_speed < stopping_speed:
61           stopping_distance = safety_margin
62           slow_distance = ego_length / 2
63       else:
64           stopping_distance = ((ego_forward_speed - stopping_speed) ** 2) / (2 * 0.7)  +
     ↪   safety_margin + ego_length / 2
65           slow_distance = ((ego_forward_speed - slow_speed) ** 2) / (2 * 0.7) + safety_margin
     ↪   + ego_length / 2
66
```

```python
67      # Step 5: Check if there is a stop sign and the distance is smaller than the stopping
        ↪   distance. If yes, initiate a stop action if the speed is greater than the stopping
        ↪   speed.
68      if distance_to_stop_sign is not None and distance_to_stop_sign < stopping_distance:
69          if ego_forward_speed <= stopping_speed:
70              speed_level = "MOVE"
71          else:
72              stopping_speed_current = max(distance_to_stop_sign / 2, stopping_speed)
73              if ego_forward_speed > stopping_speed_current:
74                  ego_forward_speed = stopping_speed_current
75                  speed_level = "STOP"
76              else:
77                  speed_level = "MOVE"
78
79      # Step 6: Check if there is a red light and the distance is smaller than the stopping
        ↪   distance. If yes, initiate a stop action.
80      elif distance_to_red_light is not None and distance_to_red_light < stopping_distance:
81          speed_level = "STOP"
82
83      # Step 7: Check for vehicles and pedestrians that may cause collision course, and
        ↪   decide whether to STOP, SLOW or MOVE the ego vehicle.
84      else:
85          collision_vehicle = False
86          collision_pedestrian = False
87          min_longitudinal_distance = stopping_distance
88          min_lateral_distance = float('inf')
89          for k, v in vehicles_rel.items():
90              location_rel = v['location_rel']
91              speed = v['speed']
92              length = v['length']
93              width = v['width']
94              longitudinal_distance = location_rel[0] - v['length'] / 2 - ego_length / 2
95
96              # check if there is a collision course with the ego vehicle
97              if np.abs(location_rel[1]) <= width / 2 + ego_width / 2 and
                ↪   longitudinal_distance <= stopping_distance:
98                  collision_vehicle = True
99                  if longitudinal_distance <= 0:
100                     speed_level = "STOP"
101                     break
102             # check if the vehicle is within safety margin
103             if longitudinal_distance < stopping_distance and np.abs(location_rel[1]) <=
                ↪   width / 2 + ego_width / 2:
104                 if np.abs(speed - ego_forward_speed) < 0.5 and speed <= ego_forward_speed:
105                     continue
106                 min_longitudinal_distance = min(longitudinal_distance - v['length'] / 2 -
                    ↪   ego_length / 2, min_longitudinal_distance)
107                 min_lateral_distance = np.minimum(width / 2 + ego_width / 2 -
                    ↪   np.abs(location_rel[1]), min_lateral_distance)
108
109         for k, v in pedestrians_rel.items():
110             location_rel = v['location_rel']
111             speed = v['speed']
112             length = v['length']
113             width = v['width']
114             longitudinal_distance = location_rel[0] - length / 2 - ego_length / 2
115
116             # check if there is a collision course with the ego vehicle
117             if np.abs(location_rel[1]) <= width / 2 + ego_width / 2 and
                ↪   longitudinal_distance <= stopping_distance:
118                 collision_pedestrian = True
119                 if longitudinal_distance <= 0:
120                     speed_level = "STOP"
121                     break
122             # check if the pedestrian is within safety margin
123             if longitudinal_distance < stopping_distance and np.abs(location_rel[1]) <=
                ↪   width / 2 + ego_width / 2:
124                 if np.abs(speed - ego_forward_speed) < 0.5 and speed <= ego_forward_speed:
125                     continue
126                 min_longitudinal_distance = min(longitudinal_distance - length / 2 -
                    ↪   ego_length / 2, min_longitudinal_distance)
127                 min_lateral_distance = np.minimum(width / 2 + ego_width / 2 -
                    ↪   np.abs(location_rel[1]), min_lateral_distance)
128
129         # Step 8: Initiate a stop action if the ego vehicle is about to collide with a
            ↪   nearby vehicle or pedestrian.
130         if collision_vehicle or collision_pedestrian or min_longitudinal_distance <=
            ↪   safety_margin/2 or min_lateral_distance <= safety_margin/2:
131             speed_level = "STOP"
132             ego_forward_speed = 0
```

```
133          # Step 9: Initiate a slow action if the vehicles or pedestrian within the safe
             ↪    stopping distance margin.
134          elif min_longitudinal_distance <= slow_distance and min_longitudinal_distance >=
             ↪    stopping_distance and min_lateral_distance <= ego_width:
135              speed_level = "SLOW"
136              if np.abs(min_lateral_distance) > 0 and np.abs(min_lateral_distance -
                 ↪    ego_width) > 0:
137                  speed_factor = (min_longitudinal_distance - stopping_distance) /
                     ↪    (slow_distance - stopping_distance/2)
138                  speed_factor = min(max(0.0, speed_factor), 1.0)
139                  ego_forward_speed = slow_speed * speed_factor + ego_forward_speed * (1 -
                     ↪    speed_factor)
140          # Step 10: Initiate a move action if no obstacles are present
141          else:
142              speed_level = "MOVE"
143              ego_forward_speed = min(ego_forward_speed + 0.2, 6.0)
144
145      # Step 11: Compute the angle between the ego vehicle orientation and the vector
         ↪    pointing to the target in world coordinates.
146      target_direction = ego_target_location - ego_location
147      target_direction_ego = np.dot(target_direction, ego_orientation),
         ↪    np.dot([-ego_orientation[1], ego_orientation[0]], target_direction)
148
149      # Step 12: Rotate the vector to the coordinate system of the ego vehicle and return the
         ↪    angle.
150      target_angle = np.arctan2(target_direction_ego[1], target_direction_ego[0]) * 180.0 /
         ↪    np.pi if np.linalg.norm(target_direction_ego) > 0 else 0.0
151      target_angle = ((target_angle + 180) % 360) - 180
152
153      return speed_level, target_angle
```

Listing A.7: Example driving policy generated by LangProp, trained with both imitation learning and reinforcement learning.

## F    FUTURE WORK

LangProp is a framework that harnesses the capability of LLMs to apply data-driven optimization techniques to code optimization. We do not claim that a solution using LangProp is appropriate for all problems - in fact, neural networks excel in working with continuous state-action spaces and low-level control, whereas LLMs have advantages in handling high-level planning and reasoning tasks, rather than low-level control tasks. Our intention is to propose an alternative learning paradigm that allows LLMs to be used to learn high-level planning which has hitherto been a difficult problem for other machine learning approaches (e.g. neural networks).

There are numerous future research directions that could improve the capability of LangProp as a training framework, as well as give a better theoretical foundation, such as (a) chaining of modules with a full back-propagation algorithm, (b) improvements to the evolutionary algorithm (e.g. priority mechanism), (c) a robust sampling mechanism for failed examples upon updates, (d) incorporating human feedback in natural language during policy updates, and (e) using LangProp with LLMs fine-tuned for code correction and optimization tasks. In particular, scaling our approach to larger repositories and complex systems would require a multi-modular approach that can propagate useful learning signals to subcomponents if there are multiple failure points in the system.

Applying LangProp to reinforcement learning tasks has open questions in credit assignment and value estimation. We have demonstrated that reinforcement learning policies written as code can be improved using LangProp if either (a) the policy can be optimized on episodic returns with a Monte-Carlo method (e.g. CartPole), or (b) there is immediate feedback from the environment (e.g. infractions in CARLA). However, for complex tasks that have delayed rewards, it is necessary to have an accurate value/advantage estimator for credit assignment. Since replacing a neural value estimator with a code-based function is not feasible, it is most likely that a hybrid method (having an interpretable code-based actor policy trained with LangProp that uses a value function estimated by a neural network as a critic) would be a way to apply LangProp to complex reinforcement learning scenarios. However, this is also an open-ended question, which calls for further exploration.

Having an LLM in the RL optimization means that we could potentially harvest more useful signals from the environment, rather than relying just on sparse scalar rewards for updates. For instance,

having descriptive feedback from the Gymnasium environment on the failure modes of the agent, given either as a warning or natural language feedback, can significantly accelerate the learning of the RL agent. This also allows a more seamless integration of human-in-the-loop feedback.

Finally, more investigation is required in terms of the robustness and safety of LLM-written applications. This is applicable to all systems that involve code generation. While our framework iteratively improves the quality of the code and filters out potential errors that make the final code policy less likely to contain errors, additional safety mechanisms and firewalls are necessary during the training process, since the code is evaluated based on execution, which could potentially be a source of attacks or risk. We stress the importance of additional safety precautions before deployment.

We believe that LangProp opens up new possibilities for data-driven code development. While zero-shot applications of LLMs have enabled tools such as GitHub Copilot, some suggestions are inaccurate or misaligned with the user's intentions, whereas if we have data or unit tests that the code needs to satisfy, the code suggestions can be made much more accurate by first running evaluations on these test suites and choosing the best possible suggestion that satisfies the requirements. Planning is one aspect of autonomous driving that has not yet successfully adopted a data-driven approach, for good reasons, since neural networks often struggle to produce generalizable high-level planning rules and are less interpretable. Therefore, most methods currently in deployment have human-engineered planning algorithms. Our LangProp framework is far from sufficient to replace such systems since it lacks the robustness that human-designed systems have to offer, and more research needs to be done in this direction. We hope that our work will provide inspiration for future research to make the framework more robust and safely deployable in the real world.

