# OpenReview forum: "LangProp: A code optimization framework using Language Models applied to driving"
_ICLR.cc/2024/Conference — Submitted to ICLR 2024_

### Official Review · Reviewer_ayZc · 2023-10-17

**Soundness:** 2 fair
**Presentation:** 2 fair
**Contribution:** 2 fair
**Rating:** 3
**Confidence:** 4

**Summary:**

This paper proposes to adopt LLM to generate codes for the planning part of the autonomous driving. The planning code could be updated according to certain scores (IL/Dagger/RL) with prompts feeding into LLM.

**Strengths:**

1. The idea of generate planning code by LLM is interesting.

**Weaknesses:**

1. After complex training pipelines (online IL + RL), the generated code's performance is still far from hand-crafted rules which has only hundreds of lines (55 vs 76 in longest6). It casts doubts on the effectiveness of the proposed method.

2. The motivation is unclear to me. The authors assume that the perception problem is solved since they feed ground-truth information about the scene to LLM.  The experiments is conducted in a further simplified environment only including ego states, other agents' states, traffic lights, and stop signs. However, in the actual driving scenario, there could lots of complex obstacles or information should be noticed in the driving scene, which might not be easy to describe precisely in language. (It is note worthy that 3D occupancy and open-set scene understanding are hot topics discussed in the community). Thus, I am not sure this framework is particularly useful for the actual usage of the planning process.

3. Novelty. The authors claim that *we are the first to completely translate and apply the training paradigm used in machine learning for iterative code generation.* However, as mentioned by the authors, works like Auto-GPT and MineDojo VOYAGER also have similar paradigms. My research focus is on autonomous driving and I am not an expert in LLM.  Thus, I am not sure whether it is an overclaim. I hope the author could explain that how your paradigm differs significantly from existing LLM Agent works instead of an implementation in the autonomous driving field.

In summary, while I appreciate the authors' engineering efforts, I do not think it is particularly useful for the actual usage of autonomous driving. Besides, the model's performance is actually limited (< hundreds of lines of hand-crafted code). Thus, I give a Reject. However, the authors are encouraged to explain the novelty and contribution of LangProp in the LLM+Code area (beyond autonomous driving) based on your claim *we are the first to completely translate and apply the training paradigm used in machine learning for iterative code generation*. I might change my score accordingly.

**Questions:**

See weakness part.

For Table 1 - LangProp: Offline IL, it seems that the agent almost did not move at all (R~=0). I am wondering what code scripts cause that and why?

---

> ### Author Response · Authors · 2023-11-15
> **Authors’ response [outlining the advantages of LangProp as a learning paradigm, differences with previous methods and the value of the research contribution]**
>
> Thank you very much for reviewing the paper. We appreciate your input as an expert in autonomous driving, and your second point of feedback is especially insightful. We would like to share with you our insight as well, and why we believe that our research in LangProp may lead to some exciting breakthroughs down the road. (While we do not want to overclaim our research achievements, we haven’t seen the paradigm shift of directly optimizing code trainable “models” in other literature, and believe that our method may open up some new avenues of research.)
>
> 1. After complex training pipelines (online IL + RL), the generated code's performance is still far from hand-crafted rules which has only hundreds of lines (55 vs 76 in longest6). It casts doubts on the effectiveness of the proposed method.
>
> While this is a valid criticism, we would like to point out that the results are still remarkable, considering that researchers (including ourselves) have invested a lot of time in optimizing towards the CARLA driving benchmark with the given scenarios, with manual tuning of hyperparameters of the driving policy, whereas the LangProp agent was able to learn a sensible driving policy from imitation learning and infraction penalties, which outperformed many of the handcrafted experts implemented in previous work. In fact, the only expert that we do not outperform is TF++ and our own expert agent. While we acknowledge there is still room for improvement, we do not think this is the shortcoming of the LangProp method itself, but because of its novel approach which still has room for exploration and refinement in future work. The other benefit of the LangProp approach is that, because of its data-driven nature, it could quickly adapt to changes in the assumptions made in the training scenarios. With manually implemented agents, changes in scenarios or assumptions may break the implementation completely, whereas with LangProp one could improve and adapt the driving policy to new sets of assumptions, as long as there is training data for imitation learning and infraction signals provided.
>
> 2. The authors assume that the perception problem is solved since they feed ground-truth information about the scene to LLM. The experiments is conducted in a further simplified environment only including ego states, other agents' states, traffic lights, and stop signs. However, in the actual driving scenario, there could be lots of complex obstacles or information should be noticed in the driving scene, which might not be easy to describe precisely in language.
>
> This is a great observation. This is exactly why we suggest the use of LLMs to generate driving policies as “code”, as opposed to using LLMs for scene description in language space, or doing language-to-actions. As you correctly point out, there are many details in the scene that cannot be captured by language, because language does not have the same granularity and spatial resolution as images or other geometric measurements. What we are proposing here is not about describing the scene in language (which, funnily enough, was what we initially considered before we came to the same conclusion as what you have pointed out just now), but using LLMs to refine driving code. While we demonstrated the capabilities of generating driving policies from scratch in our experiments, it is also possible to initialize the driving policy with human-written code, and refine them using the LangProp framework. It is true that the current LLMs are still not as robust as we would like them to be. LangProp is one way of addressing the shortcomings of its generation capabilities, and ensuring that the code quality improves with more training in a data-driven way. As to why we focus more on privileged information problems rather than directly working with RGB-D inputs, this is because “code” has an advantage over neural networks in terms of high-level reasoning and planning, which is where we want to apply a data-driven approach. Neural networks are more capable of handling continuous state-action spaces and low-level control, so rather than replacing these with LLMs, we consider a hybrid scenario where we use these neural network modules to process and extract information about the environment, and then apply the LangProp method to optimize the high-level planner.

---

> ### Author Response · Authors · 2023-11-15
> **Authors' response [continued]**
>
> 3. Novelty. The authors claim that we are the first to completely translate and apply the training paradigm used in machine learning for iterative code generation. However, as mentioned by the authors, works like Auto-GPT and MineDojo VOYAGER also have similar paradigms. I hope the author could explain how your paradigm differs significantly from existing LLM Agent works instead of an implementation in the autonomous driving field.
>
> Thank you for this opportunity to clarify our contribution. Both Auto-GPT and Voyager are different in the approaches that they take from ours, and so far, we have not come across work that is similar to LangProp, which extends the paradigm of machine learning to treating “code” as “models that can be optimized”. With this paradigm shift, we are now able to apply classical machine learning techniques (e.g. supervised learning, reinforcement learning) for data-driven code improvement. This broadens the definition of what we consider “learnable” and “trainable”, and can open up many new avenues of research that integrate learning methods into high-level reasoning, planning, and rule-based systems. We won’t repeat the arguments we have made in the paper in favor of the novelty of our method, but we will highlight the differences between our method and existing methods such as Auto-GPT and Voyager.
>
> Auto-GPT is relevant as an embodied agent system, but is a very different paradigm from LangProp. Firstly, it is not a learning paradigm, since what Auto-GPT does is to create a plan to achieve its goal, and execute the plan in finer and finer detail. While it does interact with the environment and use the responses and feedback for executing its next steps, fundamentally its planning capabilities are upper-bounded by the capability of the LLM to create plans in the first place, because there isn’t any further learning signal that refines the plan. Therefore, Auto-GPT is more akin to using the LLM as a pre-trained policy and executing it in the real world (which works surprisingly well). Secondly, Auto-GPT does not have the notion of generating code and executing it. Rather, it sequentially calls APIs and takes other pre-assigned actions.
>
> Voyager, while more similar to LangProp and has provided the inspiration for our research, is also very different in their approach. Firstly, how they solve Minecraft is more similar to how Auto-GPT solves real-world problems in that they assume that the LLM knows enough about Minecraft such that it can create a step-by-step decomposition of the plan into “skills”, and write code for each skill separately. The implementation and learning method of Voyager is tightly coupled with Minecraft (more specifically, the Mineflayer bot interface), and assumes details specific to Minecraft (e.g. inventory, health, food, equipment, crafting table). Further, as discussed in the literature review, there is no mechanism to optimize or remove a sub-optimal skill once it has been added to the library.
>
> LangProp is a method that jointly optimizes a code policy towards an objective defined by the user, provided a training dataset. This is in contrast to previous work which does not adopt the machine learning paradigm, i.e. a policy should be learnable from training data given a training objective. With our paradigm, we can extend the machine learning paradigm not only to parameterized neural networks, but also to “code”, which is a better medium to express complex logic, reasoning and planning in an interpretable way. As far as we know, we are the first to propose this way of thinking about code optimization, and treating code as checkpoints just like neural network parameters, simply because the ability of LLMs to optimize code only improved to a practically usable level in the last two years.

---

> ### Author Response · Authors · 2023-11-15
> **Authors' response [continued]**
>
> 4. For Table 1 - LangProp: Offline IL, it seems that the agent almost did not move at all (R~=0). I am wondering what code scripts cause that and why?
>
> This is an example that demonstrates the interpretability of our LangProp approach of “using code as models”, rather than having a black-box neural network as a driving policy that handles high-level planning. Causal confusion is a known phenomenon in autonomous driving where just training a neural network model on imitation learning datasets results in a driving policy that just copies the previous velocity (for instance, it is more likely that the vehicle will remain still if it is already still). This is a shortcut that models tend to make because this is a much simpler and easy-to-learn solution than a more complex policy. This phenomenon has been studied in the context of training neural network driving policies [1], but we believe us to be the first to reproduce this phenomenon in the context of code generation. It is very interesting that this is a reproducible behavior, and moreover that where exactly this happens can be identified in the code (please see section 5.2). In our pre-trained checkpoint, the casual confusion can be found at https://github.com/langprop-iclr24/LangProp/blob/2cabaaa973f1bf6dc05ce1596b41867fde666ce4/checkpoints/offline_il_(causally_confused)/predict_speed_and_steering/record_0.txt#L159.
>
> 5. The novelty and contribution of LangProp in the LLM+Code area (beyond autonomous driving) based on your claim “we are the first to completely translate and apply the training paradigm used in machine learning for iterative code generation”
>
> We are considering adding more examples of using LangProp for problems apart from driving since that was a request made by multiple reviewers. We agree that having more example use cases would be a great addition. We think it would be very interesting to apply our method to problems such as solving Sudoku or other puzzles and observe what solutions are discovered by LangProp over training. We will get back to you by the end of the rebuttal period regarding these additional examples.
>
> Finally, we thank you for your review, and hope that our response has addressed your concerns, and also clarified the novelty and contribution of our method, more from an LLM research perspective. We would appreciate a reconsideration of your review, and are happy to answer any further questions you may have.
>
> [1] Pim De Haan, et al. Causal confusion in imitation learning. Advances in Neural Information Processing Systems, 32, 2019.

---

> ### Author Response · Authors · 2023-11-22
> **Authors' response [Updated the paper reflecting your feedback]**
>
> Thank you again for your review.
>
> We have now added additional examples featuring Sudoku and CartPole to the repository. Please refer to Appendix A.9, as well as https://github.com/langprop-iclr24/LangProp/tree/main/src/langprop#more-examples for the details on the experiments to show the generality of the framework. We would like to highlight that both experiments required no modifications to the pre-existing code-base, and that the training code for the new experiments is kept minimal and can be applied to other tasks. We have also added examples of checkpoints, prompts, initially obtained solutions and final solutions after training with LangProp. In particular, we would like to point you to the CartPole example, where the initial policy (https://github.com/langprop-iclr24/LangProp/blob/main/src/langprop/examples/cartpole/example_checkpoint/zero_shot_policy.txt) learned a simplistic policy and only obtained a score of 9.9, but the trained policy (https://github.com/langprop-iclr24/LangProp/blob/main/src/langprop/examples/cartpole/example_checkpoint/trained_policy.txt) refined its strategy to include a PID controller, obtaining a score of 500 (maximum).
>
> We acknowledge that some more complex scenarios may require further investigation, possibly requiring a hybrid approach of both an interpretable code-based policy and a neural network-based value estimator. We expanded our discussion in the future work section in Appendix E, where we outline the characteristics of the current LangProp framework and how this may be improved.
>
> Thank you very much for helping us strengthen the paper, hopefully with more transparency.

---

### Official Review · Reviewer_woM2 · 2023-10-30

**Soundness:** 3 good
**Presentation:** 3 good
**Contribution:** 3 good
**Rating:** 6
**Confidence:** 3

**Summary:**

In this paper, the authors propose a code generated framework leveraged by LLMs, named LangProp. This framework can generate interpretable and transparent code that can be verified and improved in a metric- and data-driven way. The authors demonstrate the first proof of concept of automated code optimization for autonomous driving in CARLA.

**Strengths:**

1.	The research topic is innovative and carries significant relevance.
2.	The proposed LangProp is a code generation methodology that leverages the power of LLM to generate syntactically and logically compliant code.
3.	The proposed LangProp demonstrates the first proof of automated code optimization in the field of autonomous driving, generating interpretable and transparent driving strategies that can cope with some complex scenarios in CARLA.

**Weaknesses:**

1.	The framework's capacity to deliver driving strategies that are both interpretable and transparent raises concerns.
2.	The framework depends on the quality and coverage of LLM, which limits the effect and performance of LangProp. If LLM fails to generate sensible initial code, or struggles with some syntactic or logical issues, then LangProp is also hard to optimize. Existing LLMs may face some generation challenges or high uncertainty.
3.	Further analysis and in-depth exploration of the suitability of LLM in the context of autonomous driving is essential.

**Questions:**

My main concerns and questions lie in the weaknesses. The author should discuss them in detail.

---

> ### Author Response · Authors · 2023-11-15
> **Authors' response [Our primary contribution is in the LLM domain, in which LangProp strictly improves performance over zero-shot uses of LLMs. We fully agree that future work in this research area is required before safe deployment.]**
>
> Thank you very much for taking the time to review the paper. We would like to address your questions. Also, please see our responses to some of the other reviewers since this might also address some of your concerns.
>
> 1. The framework's capacity to deliver driving strategies that are both interpretable and transparent raises concerns.
>
> Since some of the other reviewers missed this, we would like to highlight that we provide checkpoints (the policy code generated by LLM using the LangProp framework) at https://github.com/langprop-iclr24/LangProp/tree/main/checkpoints. As mentioned in the paper, “online_il_rl_(best)” provides the best checkpoint which was trained using both imitation learning and reinforcement learning losses, and record_0.txt contains the best-performing policy. By interpretability, we mean that the policies that are learnt are directly inspectable and diagnosable since they are in code form. Please note that we do not claim that LangProp can replace neural network models - rather, we see our method to be complementary to more classical machine learning approaches, since by leveraging the capability of LLMs whose strengths are in high-level reasoning and planning, we can use them in combination with deep learning models which excel in working with continuous state-action spaces and low-level control to solve problems which were previously difficult to solve using a learning-based approach (e.g. planning and reasoning).
>
> 2. The framework depends on the quality and coverage of LLM, which limits the effect and performance of LangProp. If LLM fails to generate sensible initial code, or struggles with some syntactic or logical issues, then LangProp is also hard to optimize. Existing LLMs may face some generation challenges or high uncertainty.
>
> While having a bad initialization of policies is not a problem for our method, since such policies will be discarded during training as better policies emerge, the diversity of responses (tunable by the temperature parameter for the LLM), as well as the number of responses per query, are beneficial to exploring the solution space, thus reaching a feasible solution quickly. It is true that the LLM method does not solve all problems and there may be some problems that are inherently challenging for LLMs to solve. We do not claim that our method can solve all problems - in fact, there are certain classes of problems (e.g. low-level control) for which neural networks are better candidate solutions. LangProp’s contribution is in that we improve the capabilities of LLMs by a large amount by introducing this notion of optimizing the output solution towards the target objective. Without this (zero-shot prediction from an LLM), the generated code may not be aligned with the intention of the user or the task specification, as we found when we generated driving policies either zero-shot or trained on offline imitation learning. A fair assessment of the effectiveness of our method would be to compare zero-shot performance from LLMs and performance after training using LangProp.
>
> 3. Further analysis and in-depth exploration of the suitability of LLM in the context of autonomous driving is essential.
>
> We believe that what LangProp proposes (applying LLMs to optimize code in a data-driven way) opens up new possibilities for data-driven code development. While zero-shot applications of LLMs have enabled tools such as GitHub Co-pilot, some suggestions are inaccurate or misaligned with the user’s intentions, whereas if we have data or unit tests that we need to satisfy, the code suggestions can be made much more accurate by first running evaluations on these test cases and choosing the best possible suggestion that satisfies the requirement. We do not imply that autonomous driving is fully solvable with our method. In fact, as we have mentioned above, certain aspects of autonomous driving, such as perception and low-level control are best handled with neural network models, whereas our LangProp method has an advantage in iteratively optimizing code for high-level planning, which can be tested in simulation. Planning is one aspect of autonomous driving that has not yet successfully adopted a data-driven approach, for good reasons, since neural networks often struggle to produce generalizable high-level planning rules and are less interpretable. Therefore, most methods currently in deployment have human-engineered planning algorithms. Our LangProp framework is far from sufficient to replace such systems since it lacks the robustness that human-designed systems have to offer, and more research needs to be done in this direction, since we are the first to propose this approach. We hope that our work will provide inspiration for future research to make the framework more robust and safely deployable in the real world.
>
> We thank you for your review and hope that our answers have addressed your concerns and questions. We are happy to answer any further questions you may have.

---

> ### Author Response · Authors · 2023-11-22
> **Authors' response [Updated the paper reflecting your feedback]**
>
> Thank you again for your review. We found your feedback regarding safety and robustness concerns to be helpful, and have expanded our discussion on future work in Appendix E, where we stress the importance of safe deployment with additional risk mitigation mechanisms, potential strengths and weaknesses of LLM-based methods, as well as areas of investigation in future work.
>
> We have added additional examples featuring Sudoku and CartPole to the repository. Please refer to Appendix A.9, as well as https://github.com/langprop-iclr24/LangProp/tree/main/src/langprop#more-examples for the details on the experiments to show the generality of the framework. We would like to highlight that both experiments required no modifications to the pre-existing code-base, and that the training code for the new experiments is kept minimal and can be applied to other tasks. We have also added examples of checkpoints, prompts, initially obtained solutions and final solutions after training with LangProp. In particular, we would like to point you to the CartPole example, where the initial policy (https://github.com/langprop-iclr24/LangProp/blob/main/src/langprop/examples/cartpole/example_checkpoint/zero_shot_policy.txt) learned a simplistic policy and only obtained a score of 9.9, but the trained policy (https://github.com/langprop-iclr24/LangProp/blob/main/src/langprop/examples/cartpole/example_checkpoint/trained_policy.txt) refined its strategy to include a PID controller, obtaining a score of 500 (maximum).
>
> We acknowledge that some more complex scenarios may require further investigation, possibly requiring a hybrid approach of both an interpretable code-based policy and a neural network-based value estimator. We also discuss this in Appendix E, where we outline the characteristics of the current LangProp framework and how this may be improved.
>
> Thank you very much for helping us strengthen the paper, hopefully with more transparency.

---

### Official Review · Reviewer_cL61 · 2023-10-31

**Soundness:** 3 good
**Presentation:** 3 good
**Contribution:** 4 excellent
**Rating:** 8
**Confidence:** 3

**Summary:**

The paper proposed a framework, LangProp, to produce good performing codes by iteratively refining them by an LLM with execution feedback. LangProp works as a classic machine learning training process conceptually, however, the model under training is a piece of code, the reward / feedback is the execution result of the code, and the optimizer is an LLM that modifies the code based on the reward and feedback. A number of machine learning techniques can be applied to this learning process as well, such as online/offline learning, DAgger and RL.

To demonstrate the viability and performance, the paper applied LangProp to an self-driving cars problem, where the model to be optimized is a python program that drives a car (in simulated environment, CARLA). Experiments showed that the training process converges and LangProp can produce a reasonably good program that drives the car. The paper also compared the performances of different ML "meta"-techniques (DAgger/RL); online DAgger/RL has the best performance.

The authors will open source the data and source code of LangProp.

**Strengths:**

- Novelty. This is an innovative use of LLM in iterative code refinement. There is previous work on how LLM can help with debugging and code refinement, but the Reviewer has not seen the use of LLM as an optimizer in a meta learning setup. (It's totally possible that Reviewer missed some previous work)

- Effectiveness. Experiments showed that the meta-learning process actually works and the training will converge. This is a good signal.

**Weaknesses:**

- Lack of variety in experiments. To claim that LangProp is a viable meta-learning framework, we need some more examples than driving. The driving problem in the paper is more like estimating a driving function ("our expert") by a piece of python code. It's practical and useful, but also special. The Reviewer loves to see a few more different applications, especially if the optimization phase (code refinement) will always converge.

- No examples of the resulting code (policy). One of the claimed advantages is that a learned piece of code is more understandable than deep model weights. However in this optimization, it's possible that the code ends up being unfriendly to human. It will be very helpful to show some examples of good and bad performing policies in the paper.

- High cost. The training process will resulted in many calls to an LLM which could be costly. It will be helpful if the paper can offer a performance/cost curve to help decide the practicality for certain tasks.

Note: Reviewer hesitates to accept the paper as-is (with only writing improvements), unless convincing examples of learned policy are available. One or more applications would be a huge plus.

Update after rebuttal: Reviewer saw the example policy in the github project, they are indeed understandable by human. I see the points other rejecting reviewers made, but will still maintain the accept suggestion mostly based on the new way of doing things, not on the cost/effectiveness/performance of the resulted policy.

**Questions:**

- Section 1 paragraph 2. The second sentence is super long, consider break it down for easy reading.

- Section 3.1.5 Paragraph 1. "For each of these policies, the tracker is queried..." what is the tracker? Better explain to reduce reader's mental workload.

- Reviewer suggests removing "applied to driving" from the title since this is about ML, not driving.

---

> ### Author Response · Authors · 2023-11-15
> **Authors’ response [thank you for your concrete and helpful suggestions, considering adding more examples]**
>
> Thank you very much for taking the time to review the paper. We thank you for your concrete and helpful suggestions in your questions section, and are delighted that you appreciate the novelty in our work which we are also excited about.
>
> With regards to your point on a lack of variety in experiments, this is a valid point which we would like to consider. We initially had a larger focus on the autonomous vehicle aspect of the research before we realized that the framework could be made more generally available as a learning framework. We agree that having more example use cases would be a great addition. While we consider setting up experiments for e.g. Minecraft to be on the scale of a separate research project (please also see our response to Reviewer ZuhC), we think it would be very interesting to apply our method to problems such as solving Sudoku or other puzzles and observe what solutions are discovered by LangProp over training. We will get back to you by the end of the rebuttal period regarding these additional examples.
>
> In terms of examples of the resulting code (policy), these are already available in our open-sourced code (https://github.com/langprop-iclr24/LangProp/) that we have submitted along with the paper. We provide checkpoints (the policy code generated by LLM using the LangProp framework) at https://github.com/langprop-iclr24/LangProp/tree/main/checkpoints. As mentioned in the paper, “online_il_rl_(best)” provides the best checkpoint which was trained using both imitation learning and reinforcement learning losses, and record_0.txt contains the best-performing policy. The offline trained policy is the one with causal confusion (setting the future speed to zero when the vehicle is already stationary), which we found interesting to be able to diagnose just by inspecting the human-readable code. It seems that some of the other reviewers also missed this, so we apologize if this wasn’t made clear, and will add further description to the reproducibility statement highlighting this.
>
> Regarding the cost, we did mention the cost per job in Appendix C.3, however, the estimate we provided turned out to be an underestimate once we actually received the billing (due to underestimating the number of tokens per response), so we will correct it to $150 per job. This is substantial but not significantly high in terms of training costs in ML. Also, OpenAI recently reduced their API costs (as well as increasing the token limit), so the costs will further decrease in future. The cost is dependent on several factors, namely the number of responses per query (in our case 2), the number of policies that are updated per every training step (in our case 2), and the frequency of updates (in our case once per 100 batch samples). Unfortunately, every training job takes 5-6 days to complete (this bottleneck comes both from GPT 3.5’s response time, as well as the speed of the CARLA simulator which takes 2-3 days for a full training route run even without any queries to GPT 3.5), so we weren’t able to create a cost/performance curve.
>
> Thank you for your suggestions in the questions section. We have integrated your first comments and will update the document accordingly. In terms of the title change, we agree that LangProp is a more general framework than applications to autonomous driving, but since our main experiments are in showing results in CARLA (and we also would like to highlight our research contribution in the CARLA domain in terms of providing a high-performing expert baseline), we might opt for maintaining the current title.
>
> Finally, we would like to thank you once again for your many concrete and constructive suggestions, and appreciate your positive feedback. We will try to implement some additional example use cases of LangProp and will get back to you before the end of the rebuttal period.

---

> > ### Comment · Reviewer_cL61 · 2023-11-22
> > **Thanks for all the answers!**
> >
> > Thanks for the detailed answers. I think the lack of more usecases is a weakening point. I checked the GitHub code examples: they are indeed understandable by human.

---

> ### Author Response · Authors · 2023-11-22
> **Authors' response [Updated the paper reflecting your feedback]**
>
> Thank you very much! We have now updated the paper, reflecting your comments on Section 1 paragraph 2. and Section 3.1.5 Paragraph 1.
>
> We have also added additional examples featuring Sudoku and CartPole to the repository. Please refer to Appendix A.9, as well as https://github.com/langprop-iclr24/LangProp/tree/main/src/langprop#more-examples for the details on the experiments to show the generality of the framework. We would like to highlight that both experiments required no modifications to the pre-existing code-base, and that the training code for the new experiments is kept minimal and can be applied to other tasks. We have also added examples of checkpoints, prompts, initially obtained solutions and final solutions after training with LangProp. In particular, we would like to point you to the CartPole example, where the initial policy (https://github.com/langprop-iclr24/LangProp/blob/main/src/langprop/examples/cartpole/example_checkpoint/zero_shot_policy.txt) learned a simplistic policy and only obtained a score of 9.9, but the trained policy (https://github.com/langprop-iclr24/LangProp/blob/main/src/langprop/examples/cartpole/example_checkpoint/trained_policy.txt) refined its strategy to include a PID controller, obtaining a score of 500 (maximum).
>
> We acknowledge that some more complex scenarios may require further investigation, possibly requiring a hybrid approach of both an interpretable code-based policy and a neural network-based value estimator. We expanded our discussion in the future work section in Appendix E, where we outline the characteristics of the current LangProp framework and how this may be improved.
>
> Thank you very much for helping us strengthen the paper, hopefully with more transparency.

---

### Official Review · Reviewer_ZuhC · 2023-10-31

**Soundness:** 3 good
**Presentation:** 4 excellent
**Contribution:** 4 excellent
**Rating:** 6
**Confidence:** 3

**Summary:**

This paper proposes a framework that uses LLMs for data-driven code optimization and demonstrated its capability in generating driving policies in CARLA. It allows for training with different paradigms and objective functions, leading to more interpretability and transparency in machine learning.

Featured contributions:

* The LangProp framework utilizes LLMs for code optimization in generating driving policies.
* It supports various training paradigms and objective functions for improved interpretability.
* Training with LangProp provides more transparent machine learning and opens up new possibilities.

**Strengths:**

1. The overall idea and story are innovative, proposing a set of methods that allow LLM to generate code iteratively, and can utilize existing objectives for supervision, demonstrating effectiveness in the task of driving. Also, the idea of comparing LLM to an optimizer is something I find very interesting. The discovery of causal confusion in the offline imitation experiments is also very interesting, which reflects the interpretability of this method.
2. It can have a certain influence; this framework can provide more ideas and options for subsequent researchers, for example, in the directions of LLM, symbolic systems, and embodied AI.
3. The overall quality is good, with fluent writing, detailed elaboration, and the core code has been open-sourced. I believe it has a high degree of credibility.

**Weaknesses:**

1. The paper has some overclaims, and the experimental part is not sufficient enough. It states that this is a universal framework, but why are experiments only conducted in the CARLA simulator for autonomous driving scenarios? If experimental results could be obtained in other more complex and diverse environments, such as Minecraft or tasks related to robotics, it would greatly enhance the credibility of the paper.
2. You have only showcased the LLM's capability for code optimization, which is constrained by the LLM's window size. This means that inference is limited to simple-logic functions, which are far from sufficient for the more complex interactive tasks and long-tail problems in autonomous driving. Though achieving high score in CARLA, it is way far from reality. How can we extend the code generation task to be more complex and diverse? For example, similar to VISPROG[1], to complete more complex tasks through the combination of models. I would like to hear your thoughts on this matter.
3. I am quite confused with Figure 3:
 *  how is the training score calculated for Offline IL? Do you use L1/L2 for IL while using the training objected as you mentioned in Appendix C.2 for hybrid-RL? In that case they are in difference scales. If so, how do you ensure that the objectives for IL and RL are comparable, since the scales of the objectives are completely different?  Additionally, why are they distributed in the range of [-1, 1]? Did you normalize the objectives?
 * Furthermore, I suggest adding another graph showing the close-loop score versus the number of steps. This would allow for a comparison of scores across all methods and clearly show how the final close-loop score (most important) changes as the number of steps increases. Though it could be time-consuming but very sparse sampled point would be helpful, just like validation per epoch.

[1] Gupta, Tanmay, and Aniruddha Kembhavi. "Visual programming: Compositional visual reasoning without training." Proceedings of the IEEE/CVF Conference on Computer Vision and Pattern Recognition. 2023.

**Questions:**

Please refer to the weakness part.

---

> ### Author Response · Authors · 2023-11-15
> **Authors’ response [thank you for your insightful remarks and thoughtful questions, considering adding more examples]**
>
> Thank you very much for taking the time to review the paper. We are delighted that the main message of our paper came across and you see the future potential in this branch of research as much as we do. As you suggest, there are many areas and use cases that could benefit from subsequent research.
>
> With regards to your point on having more complex and diverse environments being desirable, this is a valid point which we shall consider seriously (more action points in paragraph 4). There is a subtle difference in claiming that our framework is a general learning framework similar to PyTorch lightning (which we claim), and that it can solve all machine learning tasks (which we do not claim) - for instance, LLMs are better at handling high-level planning or reasoning tasks, rather than low-level control tasks, and our intention for this paper is to propose an alternative learning paradigm that allows LLMs to be used to learn high-level planning which has hitherto been a difficult problem for other machine learning approaches (e.g. neural networks). On the other hand, neural networks excel in working with continuous state-action spaces and low-level control. We could add this disclaimer to the paper to prevent any unwanted overclaims.
>
> As for testing this in Minecraft, this would be indeed an exciting application for our LangProp method, and since Voyager also harnesses LLMs, we believe this to be a feasible application. However, there are many specific assumptions made in Voyager that use the domain knowledge of Minecraft, so disentangling these assumptions from the learning paradigm is a non-trivial task. For our autonomous driving experiment, setting up the experiment (including the evaluation benchmark, the baselines, determining what inputs and outputs should be given, and designing the rollout buffer and reinforcement learning pipeline) was a substantial engineering effort, and we expect setting up experiments for Minecraft would be a similar endeavor, hence we considered this to be out of scope. This is not necessarily because LangProp isn’t applicable to these tasks, but because more work is required in designing RL-specific training strategies, and these RL environments do not implement LLM-friendly interfaces ready for plug and play as of now. This is similar to the era in RL research before OpenAI gym interfaces with standardized state and action spaces were defined, and one had to implement their own custom environments or adapt their RL algorithms to work with a particular environment. LangProp is a framework more akin to PyTorch lightning rather than an RL framework (e.g. Stable Baselines / RLLib).
>
> Having said this, we are keen to add more experiments showcasing the potential diverse use of LangProp (which does not require starting a new project). For example, we are considering adding examples in Sudoku and/or other similar puzzles. We hypothesize there may be some interesting emergent behavior in terms of what solutions are found if we change the learning objective (for example, we could penalize computational time if we want a more computationally efficient solution). We will update you closer to the end of the rebuttal period in terms of this.

---

> ### Author Response · Authors · 2023-11-15
> **Authors' response [continued]**
>
> Your point on the code optimization constrained to the window size is also a very good point. We have also thought about this, and we have had some trial experiments with multiple modular components (similar to the VISPROG example you’ve raised), but we found that this degraded the performance due to each of the modular components is also learnable and may not accurately match the docstring description. Rather than having a complex system which treats this problem separately, we found that including all the source code in one query and jointly optimizing the code tend to have more stable performances. However, we also believe that this multi-modular approach still has potential and is solvable with more research in this direction and ways of propagating learning signals to each of these modular components. (Please see our discussion on this topic in Appendix A.9).
>
> We do not consider the limitation due to context window size to be a significant drawback, since our framework can be used with any LLM, and there have already been rapid advances in the capabilities of LLMs. For instance, when we conducted our research we used a GPT 3.5 model with a context of 16k tokens, but now there are models that have 128k tokens with cheaper pricing. There is also a lot of research in vector embedding databases to make large repositories that far exceed the context size of LLMs searchable and usable. While there are still drawbacks that you have pointed out, we believe these concerns will be alleviated as the infrastructure around LLMs becomes more mature.
>
> Thank you for pointing out the need to clarify Figure 3. Indeed, we use the equation in Appendix C.2. The first term is the imitation learning objective (imitate the expert unless there is an infraction, or if the expert’s action is different from the infractious behavior policy), the second term is the infraction penalty (if there is an infraction and the current policy took the same action as the infractious behavior policy), and the third term is if the steering angle has an error with the ground truth angle by more than a given threshold. As you point out, the scales are different so we have coefficients r_infrac and r_angle to scale these penalties. Since these penalties are sparse, we weigh them more (-10) compared to the imitation learning objective. This is also why the range of the training objective is beyond [0, 1], in fact, it can take the range of [-10, 1] in case there are many exceptions at the beginning of training. We will correct the caption to mention this and that we have set a ylim to the score so that we can actually see the learning curve (because if we set the range to [-10, 1] then the curve is mostly not visible).
>
> As for the close-loop score versus the number of training steps, this would be ideal, however, the close-loop evaluation for each saved checkpoint in CARLA takes about 2-3 days for the training routes, 1-2 days for the test routes, and 6-7 days for the Longest6 routes, so plotting a curve is not very feasible. This is why we have a table showing the close-loop evaluation of the final checkpoint instead.
>
> Finally, we would like to thank you again for your detailed analysis of the paper and we appreciate all your insightful comments, questions and suggestions. We will get back to you in terms of adding additional examples, potentially for Sudoku and some other puzzles.

---

> ### Author Response · Authors · 2023-11-22
> **Authors' response [Updated the paper reflecting your feedback]**
>
> Thank you very much again for your feedback.
> We have updated the paper with a more descriptive caption on the range of the score in Figure 3.
>
> We have also now added additional examples featuring Sudoku and CartPole to the repository. Please refer to Appendix A.9, as well as https://github.com/langprop-iclr24/LangProp/tree/main/src/langprop#more-examples for the details on the experiments to show the generality of the framework. We would like to highlight that both experiments required no modifications to the pre-existing code-base, and that the training code for the new experiments is kept minimal and can be applied to other tasks. We have also added examples of checkpoints, prompts, initially obtained solutions and final solutions after training with LangProp. In particular, we would like to point you to the CartPole example, where the initial policy (https://github.com/langprop-iclr24/LangProp/blob/main/src/langprop/examples/cartpole/example_checkpoint/zero_shot_policy.txt) learned a simplistic policy and only obtained a score of 9.9, but the trained policy (https://github.com/langprop-iclr24/LangProp/blob/main/src/langprop/examples/cartpole/example_checkpoint/trained_policy.txt) refined its strategy to include a PID controller, obtaining a score of 500 (maximum).
>
> We acknowledge that some more complex scenarios may require further investigation, possibly requiring a hybrid approach of both an interpretable code-based policy and a neural network-based value estimator. We expanded our discussion in the future work section in Appendix E, where we outline the characteristics of the current LangProp framework and how this may be improved. We hope this will mitigate any unintended overclaims in the paper, and communicate both the potentials of LangProp and where further work is needed.
>
> Thank you very much for helping us strengthen the paper, hopefully with more transparency.

---

> > ### Comment · Reviewer_ZuhC · 2023-11-23
> > **Thank you for your detailed response**
> >
> > Thank you for your detailed response. Most of my concerns have been addressed.

---

### Official Review · Reviewer_maNz · 2023-11-03

**Soundness:** 2 fair
**Presentation:** 2 fair
**Contribution:** 2 fair
**Rating:** 6
**Confidence:** 4

**Summary:**

The paper introduces LangProp, a framework designed for the iterative optimization of code generated by large language models (LLMs) within supervised and reinforcement learning settings. LangProp addresses the suboptimal nature of zero-shot code solutions produced by LLMs, particularly their tendency to fail in edge cases. The framework enhances code by automatically evaluating its performance against a dataset of input-output pairs, identifying exceptions, and using this feedback to guide the LLM towards generating improved code iterations.

Specifically, the LangProp model consists of a setup prompt, an update prompt, and a collection of executable code generated by LLM. Given any metric of success defined by data-driven way, LangProp uses LLM to optimize policies to improve performance, with updates generated by an evolutionary algorithm. The model keeps a bunch of policies which are used for selection of the best inference model. A forward-pass is referred to as getting predictions from all policies from the inputs, and scores or exceptions are collected as feedback to LLM to improve the policies' code.

In its experiments, the paper underscores LangProp's effectiveness in producing driving policies within the CARLA simulation environment, illustrating that traditional training paradigms like imitation learning, DAgger, and reinforcement learning are compatible with LangProp. The framework's approach to code optimization, treating policy code as model parameters and using LLMs as code optimizers, brings forth a new way to guide policy selection based on performance metrics. The authors posit that LangProp fosters a new way for data-driven machine learning that offers enhanced interpretability and transparency​.

**Strengths:**

The work proposes a framework to perform code optimization based on data-driven feedback from LLM. It provides an example of applying this code optimization to the setting of autonomous driving. The results look better than baselines. The idea of using LLM to optimize the code and the implementation to apply it on imitation learning and/or reinforcement learning is interesting.

After the rebuttal, I upgrade my rating since the authors helped to clarify some of the unclear parts of the paper.

**Weaknesses:**

While the idea of using LLM to optimize code based on data-driven scores feedback is interesting, it is hard to find evidence why this is more interpretable or reliable than training with imitation learning or reinforcement learning for self-driving. LLM is still suffering from the problems of hallucinations and its output can be very stochastic and there is no existing proof that LLM can automatically solve any coding bugs. The authors only provide examples of how to use LLM to optimize a function that calculates the factorial of an integer input, but did not provide real examples of how LLM is used to optimize a code that is used for self-driving car.

This is more like a work that uses LLM to automate the process of writing imitation learning/reinforcement learning code, rather than inventing a novel way to obtain a policy for self-driving. The problems that imitation learning/reinforcement learning have will not go away using this kind of training, such as lack of interpretability, and the lack of ability of solve edge and long-tailed cases.

There is not enough detail on how the authors cope with cases where LLM could not solve bugs in the code, nor enough detail on what is the final planning policy look like for the driving task. Also it looks like the author assumed privileged access to ground truth bounding boxes, positions etc. information from the simulator, while in the real world this is always going to be some noise in these observations. It is not clear how in that case, the policies can be robust.

**Questions:**

Could you provide a concrete example code that LLM optimizes in the end that is able to drive in the CARLA environment? Is it a neural network or a rule-based code? If it is a rule-based code, how can you guarantee it will always work in any situations, or how to ensure it can generalize?

The authors claimed this is more interpretable, but as far as I can see, it is not clear how this is achieved due to the lack of examples of the final policies optimized by LLM.

---

> ### Author Response · Authors · 2023-11-15
> **Authors’ response [examples requested by the reviewer available in our open-sourced repository]**
>
> Thank you very much for taking the time to review the paper. We would like to point you to the open-sourced code (https://github.com/langprop-iclr24/LangProp/) that we have submitted along with the paper. We provide checkpoints (the policy code generated by LLM using the LangProp framework) at https://github.com/langprop-iclr24/LangProp/tree/main/checkpoints. As mentioned in the paper, “online_il_rl_(best)” provides the best checkpoint which was trained using both imitation learning and reinforcement learning losses, and record_0.txt contains the best-performing policy. It seems that some of the other reviewers also missed this, so we apologize if this wasn’t made clear, and will add further description to the reproducibility statement highlighting this.
>
> As it could be observed from the checkpoints, the policy is rule-based and is written in Python code, hence we believe it to be more interpretable over neural network policies. In terms of guarantees, it is true that our method alone would not be able to provide such provable guarantees. However, this is true with other machine learning methods as well (e.g. neural networks), and our method has an advantage over such methods in that a human expert could assess the generated code to determine if there are any bugs or deficiencies. There is a substantial amount of literature on program verification as well, which is out of scope of this paper, but could potentially significantly enhance our current method.
>
> Next, we would like to address some of the other concerns you have raised:
>
> - “it is hard to find evidence why this is more interpretable or reliable than training with imitation learning or reinforcement learning”
>
> In fact, we do use imitation learning and reinforcement learning to train our policy. The part that LangProp extends the definition of a “model” (e.g. a parameterized neural network) so that we could treat any Python function as a “model”, and apply learning frameworks such as supervised learning and reinforcement learning to improve such “models”, since this was hitherto regarded as impossible until the improvements in capabilities of LLMs.
>
>
> - “LLM is still suffering from the problems of hallucinations and its output can be very stochastic and there is no existing proof that LLM can automatically solve any coding bugs”
>
> We agree that LLMs used zero-shot generate very poor quality code which may contain many bugs and issues. Our insight was that, LLMs are actually better at diagnosing code, finding errors in code and correcting them than generating a perfect solution zero-shot. However, it requires a lot of manual labor if one were to check for errors in the code manually and prompt the LLM repeatedly. LangProp is a framework that automates the process of iterative prompting, which also allows the code to optimize towards a user-defined training objective that quantifies the success metric. Our framework is ideologically similar to neural network training, since neural networks do not have provable guarantees but can optimize its expected performance. What the user can do to minimize the probability of catastrophic failure is to assign a large penalty for such incidents so that it is probabilistically less likely for the model to exhibit such behavior.
>
> - “The authors only provide examples of how to use LLM to optimize a function that calculates the factorial of an integer input, but did not provide real examples of how LLM is used to optimize a code that is used for self-driving car.”
>
> As mentioned above, we provide examples of checkpoints from each of the experiments run on CARLA (offline imitation learning, DAgger imitation learning, DAgger imitation + reinforcement learning, online imitation + reinforcement learning).
>
> - “This is more like a work that uses LLM to automate the process of writing imitation learning/reinforcement learning code, rather than inventing a novel way to obtain a policy for self-driving.”
>
> We believe there has been some misunderstanding, since what the LLM generates is the driving policy code, not the training code for IL / RL. As for the novelty, the paradigm shift of using LLMs to optimize code (as opposed to finetuning LLMs), treating code as optimizable models is a novel proposal (at least at the time of submission).

---

> > ### Comment · Reviewer_maNz · 2023-11-23
> > **thanks for the rebuttal! I updated my comments**
> >
> > thanks for the rebuttal and answering my questions, I checked the code, and it is indeed rule-based. I'd suggest to put some of the generated code in the paper, at least in appendix. since your paper is about autonomous driving, using other examples are irrelevant. However, I do think this is still kind of limited, in that rule-based approach has been a while and it hasn't been proved to be working in production. In a complicated system, rule-based approach especially when you have multiple rules combined together would be hard to check for correctness. But the novelty is indeed there.

---

> ### Author Response · Authors · 2023-11-15
> **Authors' response [continued]**
>
> - “There is not enough detail on how the authors cope with cases where LLM could not solve bugs in the code, nor enough detail on what is the final planning policy look like for the driving task.”
>
> Please find these details in the open-sourced code repository. The final planning policies are available in the checkpoints directory.
>
> - it looks like the author assumed privileged access to ground truth bounding boxes, positions etc. information from the simulator, while in the real world this is always going to be some noise in these observations
>
> This is a fair point. However, in our evaluation of our method, we compare it against human-implemented expert agents which also have privileged access to such ground truth information (without noise), so we believe that this is a fair comparison to make. Our method outperformed most of the experts, which indicates that the code generated by LangProp is comparable to human-written code for this specific task. Note that the agents we benchmark against were developed in 2022, whereas GPT 3.5 which we use to generate code was trained on data up to September 2021, so the LLM has not seen the human-written expert solutions during training.
>
> Our focus was on showing LangProp’s capability to improve code compared to the LLM’s zero-shot performance, as well as comparing against human experts for the driving policy. We do not claim that a solution using LangProp is appropriate for all problems - in fact, neural networks excel in working with continuous state-action spaces and low-level control, whereas LLMs are better at handling high-level planning or reasoning tasks, rather than low-level control tasks. Our intention for this paper is to propose an alternative learning paradigm that allows LLMs to be used to learn high-level planning which has hitherto been a difficult problem for other machine learning approaches (e.g. neural networks). We will add this disclaimer to the paper to avoid any potential overclaims.
>
> Finally, we would like to thank you again for your time in reviewing the paper. We hope that the open-sourced repository will clarify some of the confusion, and would appreciate a reconsideration of your review.

---

> ### Author Response · Authors · 2023-11-22
> **Authors' response [Updates to the paper]**
>
> Thank you again for your review.
>
> We would like to update you that we now have added additional examples featuring Sudoku and CartPole to the repository. Please refer to Appendix A.9, as well as https://github.com/langprop-iclr24/LangProp/tree/main/src/langprop#more-examples for the details on the experiments to show the generality of the framework. We would like to highlight that both experiments required no modifications to the pre-existing code-base, and that the training code for the new experiments is kept minimal and can be applied to other tasks. We have also added examples of checkpoints, prompts, initially obtained solutions and final solutions after training with LangProp. In particular, we would like to point you to the CartPole example, where the initial policy (https://github.com/langprop-iclr24/LangProp/blob/main/src/langprop/examples/cartpole/example_checkpoint/zero_shot_policy.txt) learned a simplistic policy and only obtained a score of 9.9, but the trained policy (https://github.com/langprop-iclr24/LangProp/blob/main/src/langprop/examples/cartpole/example_checkpoint/trained_policy.txt) refined its strategy to include a PID controller, obtaining a score of 500 (maximum).
>
> We acknowledge that some more complex scenarios may require further investigation, possibly requiring a hybrid approach of both an interpretable code-based policy and a neural network-based value estimator. This is also discussed in Appendix E, where we outline the characteristics of the current LangProp framework and how this may be improved.
>
> Thank you very much for helping us strengthen the paper, hopefully with more transparency.

---

> ### Author Response · Authors · 2023-11-23
> **Thank you for your suggestion - updated the paper with generated code**
>
> Thank you very much for your updated review and for your suggestion to include the code in the appendix. We have now done so in Appendix E. It does indeed make it easier to find the code, thank you for your suggestion!
>
> With regard to your comment on the limitation, we agree that there is still a lot of scope for improvement (discussed in Appendix F Future Work) - while a rule-based approach has been thoroughly explored, we believe there is scope for substantial innovation in the space of using a hybrid approach of learnable rule-based systems combined with learnable neural systems. Combining a rule-based policy updated by a neural critic network to incorporate information on uncertainty, stochasticity and inaccuracies of the environment is just one suggestion.
>
> Thank you once again for taking your time and helping us improve our work.

---

### Official Review · Reviewer_JXcz · 2023-11-04

**Soundness:** 1 poor
**Presentation:** 2 fair
**Contribution:** 3 good
**Rating:** 1
**Confidence:** 5

**Summary:**

LangProp is a framework that utilizes Large Language Models (LLMs) to iteratively generate and optimize symbolic scripts, drawing a parallel to the way neural networks are trained. This approach combines the interpretability of Symbolic AI with the self-improving capabilities of machine learning, aiming to produce transparent systems that can improve over time through data-driven feedback. By treating code generated by LLMs as modifiable parameters, LangProp optimizes these scripts based on performance feedback, akin to traditional model training. The framework has been tested in the context of autonomous driving, demonstrating the potential to create interpretable, language-instructable systems that enhance performance with more data. LangProp's adaptability suggests it could extend beyond specific applications like Minecraft (where similar concepts have been explored) to a wide range of domains where model transparency is essential.

**Strengths:**

* proposal of a framework capable of automatically optimizing code generated by large language models.
* employs methods of supervised and reinforcement learning to autonomously evaluate the code's performance on input-output datasets and feedback the results to the language model, allowing for iterative improvement of the generated code.
* provides an example demonstrating how to use this framework to generate driving strategies for autonomous vehicles.

**Weaknesses:**

* it requires a substantial amount of training data to train the language model and optimize the generated code.
* since the framework uses methods of supervised and reinforcement learning, there may be issues with overfitting or undertraining during the training process.
* the framework requires proper normalization and standardization of inputs and outputs; otherwise, it may result in degraded performance of the generated code.

**Questions:**

1. How does the LangProp framework's evolutionary algorithm-based updating process compare in efficiency and effectiveness with the gradient-based optimization used in traditional neural network training?
2. How does the diversity of initial policies generated by the LLM affect the convergence and performance of the LangProp model in various stages of training?
3. Is the LangProp framework's priority and reranking mechanism going to be redundant when the training iteration increased?
4. How do different prompt template strategies affect the LangProp framework’s ability to generate and refine code?
5. Is there going to be a limit one generating correct code since the model parameters are not trained during the process?
6. What are the potential risks and challenges associated with the use of evolutionary algorithms for code optimization in safety-critical applications such as autonomous driving?
7. How does the framework address the issue of interpretability of the generated code?
8. What are the potential applications of the LangProp framework beyond autonomous driving, and how does it generalize to other domains and programming languages?
9. How can we quantify the trustworthiness in the generated code? Are there any ways to quantify subjectivity in driving behaviors while getting / obtaining a measure of trust in the outcome or end result of LangProp. I think the startegy in Figure 2 can gain inspiration from a trustworthiness framework developed for deep reinforcement learning in autonomous driving and CNNs:  "A general trust framework for multi-agent systems." In Proceedings of the 20th International Conference on Autonomous Agents and MultiAgent Systems, pp. 332-340. 2021 "Trustworthiness Evaluation and Trust-Aware Design of CNN Architectures." In Conference on Lifelong Learning Agents, pp. 1086-1102. PMLR, 2022 A major issue with the proposed framework is that it does not have any way of verifying and validating the safety or trustworthiness which are critical for autonomous driving.
10. How does the quality of the dataset influences the performance of the LangProp because this can depend on what driving scnearios are considered?

---

> ### Author Response · Authors · 2023-11-15
> **Authors’ response [questions appreciated, criticism generally applies to most ML methods]**
>
> Thank you for reviewing the paper. The questions that you present are insightful and show that you have taken the time to read the paper. We would like to address your questions fully in the latter half of our response.
>
> As for the weaknesses and rating, we find the criticisms raised to be unfair and lack sufficient reasoning and justification for this decision, since most of the criticism is either not representative or uniquely applicable to our method. (In fact, the comments made apply to most standard machine learning methods.) We believe there may have been some misunderstanding and would like to rectify it. Let us address each of your concerns in turn.
>
> - it requires a substantial amount of training data to train the language model and optimize the generated code.
>
> LangProp is an approach that actually requires much less domain-specific data because we use a pre-trained LLM and use its code generation capability and its open-domain generalization of language-to-code translation to solve novel tasks. Rather than finetuning the LLM, we show that iterative prompting allows for more successful outcomes compared to one-shot generation of code in a new application domain. The number of samples required for this optimization is small since we do not finetune the LLM itself. Instead, we use the LLM to optimize the code solution.
>
> - since the framework uses methods of supervised and reinforcement learning, there may be issues with overfitting or undertraining during the training process.
>
> This is true for almost all modern machine learning algorithms, and common training strategies to avoid such issues (e.g. train / val / test split, early stopping, data augmentation, training on data representative of the application domain) can be applied to training with LangProp as well.
>
> - the framework requires proper normalization and standardization of inputs and outputs; otherwise, it may result in degraded performance of the generated code.
>
> It is true that the training inputs and outputs need to be documented accurately (including the units of measurement of each quantity) in the docstring that defines whatever function you are trying to model. However, we do not see this as a deficiency of our method. In fact, models are bound to fail to perform under inputs that use a wrong normalization or unit (it is required that the user normalizes inputs to models trained on ImageNet, for example), and having such requirements written down explicitly in the docstring is better documentation, and may also be useful for automated assertions of the ranges of inputs and outputs.

---

> ### Author Response · Authors · 2023-11-15
> **Authors’ response [continued]**
>
> Next, let us address your questions.
>
> 1. How does the evolutionary algorithm-based updating process compare in efficiency and effectiveness with gradient-based optimization?
>
> The short answer is that it is not possible to apply a gradient-based update method directly in our problem setting. The longer answer is that, because we are looking at open domain adaptation of code generation (i.e. the ground truth of the optimal code is not available for a given novel task, and we optimize the code based on its accuracy), we cannot simply fine-tune and regress the language model to generate the ground truth code solution with supervised learning. Alternative approaches to our method which use gradient-based methods would be using reinforcement learning [1] or other forms of planning [2] (e.g. Monte Carlo Tree Search), but these require much more compute due to having sparse rewards (if one considers generating a token from the LLM as an action, and code as states). Furthermore, an incomplete code would fail with an exception (thus a large penalty) so it is difficult to learn an accurate value function and it is unstable to retrain an LLM using such rewards / values.
>
> Another argument against a gradient descent method is that, for our state space (i.e. code), feasible solutions may be far apart in the token representation space and is non-trivial to find one from the other by gradient descent. In contrast, an evolutionary algorithm approach works well in our use case because a mapping from one feasible solution to another (e.g. error correction) can be obtained by using an LLM, and can be used to perform updates to each seed. Once a feasible solution has been discovered, descendants from this seed are also likely to lie within the solution space.
>
> 2. How does the diversity of initial policies affect the convergence and performance?
>
> While having a bad initialization of policies is not a problem for our method, since such policies will be discarded during training as better policies emerge, the diversity of responses (tunable by the temperature parameter for the LLM), as well as the number of responses per query, are beneficial to exploring the solution space, thus reaching a feasible solution quickly. Because high-performing solutions are stored without modification in our evolutionary approach, convergence is guaranteed, as opposed to gradient-based methods with high learning rates which may result in divergence.
>
> 3. Would the LangProp framework's priority and reranking mechanism be redundant with more training iteration?
>
> The priority and reranking mechanism is used to identify and constantly improve the current best-performing policy. This is important because the best-performing policies are used for code updates, as well as for training data collection in an agent-based training setting. It is important that the agent uses the current best policy, since if the agent keeps using an old erroneous policy, it will keep making the same mistakes, for example, crashing into a vehicle in front because the agent policy hasn’t been updated.
>
> 4. How do different prompt strategies affect the LangProp framework’s ability to generate and refine code?
>
> Our research focus is orthogonal to research in prompt engineering strategies (e.g. Chain-of-Thought), since most prompt engineering strategies can still be applied in LangProp and the LangProp framework is agnostic to which prompt engineering strategy one chooses. It is true that a detailed specification of the function’s inputs and outputs as a docstring (including the units and data types of each input and output) is important. While the ablation of prompt engineering strategies is out of scope for this paper, we welcome any follow-up research on this topic.
>
> 5. Is there going to be a limit on generating correct code since the model parameters are not trained during the process?
>
> We experimented with Python code generation for tasks that can be solved by using standard Python libraries, so it is possible that the LLM either needs to be fine-tuned on an unknown programming language or be provided with documentation of a new library’s API if the task requires knowledge of languages or packages unknown to the LLM. We observe, however, that the LLM has the capability of generating code that performs better than what was in its training data. For example, we used GPT 3.5 Turbo model which was only trained on data collected up to September 2021, yet the LangProp agent outperformed human-written expert agents that were developed and published in 2022 and 2023, indicating that our method can generate code that is more performant than what was in the training data. (Note that this is not possible if we do not use LangProp, i.e. if we just use GPT 3.5 zero-shot)
>
> [1] Hung Le, et al. Coderl: Mastering code generation through pretrained models and deep reinforcement learning. NeurIPS, 2022.
> [2] Shun Zhang, et al. Planning with large language models for code generation. ICLR, 2023.

---

> ### Author Response · Authors · 2023-11-15
> **Authors' response [continued]**
>
> 6. What are the potential risks and challenges associated with the use of evolutionary algorithms for code optimization in safety-critical applications such as autonomous driving?
>
> The optimization challenge is not specific to the use of evolutionary algorithms, since as stated before, we maintain a list of high-performing policies that were generated previously and only discard low-performing policies, so catastrophic forgetting is less likely compared to gradient-based methods. The challenges with evolutionary algorithms have more to do with their scalability to larger repositories and complex systems, since the evolution criteria is based on the final accuracy of the overall system, which is not sufficient to propagate useful learning signals to subcomponents if there are multiple failure points in the system. Gradient-based methods, on the other hand, have the advantage that they can propagate denser learning signals. This limitation of LangProp is discussed in Appendix A.9. Our work does not claim to solve this multi-modular task; rather, we hope that LangProp will be a good starting point for future endeavors in this research direction.
>
> 7. How does the framework address the issue of interpretability of the generated code?
>
> The generated code is human-readable, which greatly enhances the interpretability of the system compared to a neural network model. Interpretability can also be enhanced through prompting schemes such as Chain-of-Thought during code generation, or requiring dense commenting of the code.
>
> 8. What are the potential applications of the LangProp framework beyond autonomous driving, and how does it generalize to other domains and programming languages?
>
> LangProp is highly compatible with applications that have well-defined inputs, outputs, and performance metric. Applications may include writing code for operations on spreadsheets and data structures, generating SQL queries, automated test-driven programming for backend applications, and any other task that typically benefits from automated code generation and correction. As mentioned in the response to Q5, generalization to unknown programming languages may either require an LLM trained or fine-tuned on that programming language, while generalization to a new library may only require passing in the documentation of the library API in the prompt.
>
> 9. How can we quantify the trustworthiness in the generated code?
>
> Thank you for the suggestion with regards to introducing a trust framework to the autonomous driving solution. It is true that not all the actors in the scene behave in ways typically expected from a rational agent, and hence that amount of uncertainty has to be incorporated in the model. For our LangProp framework, the way this should be incorporated would be in the success metric which dictates which policies should be prioritized. One could, for instance, have auxiliary losses for accurately predicting the actors’ future moves so that the model has some understanding of the actors’ behavior, implicitly modeling their trustworthiness.
>
> Please note that our paper’s focus is more on the learning paradigm, rather than the autonomous driving solution itself. We use CARLA as a useful standardized benchmark to demonstrate the capabilities of the LangProp learning paradigm, where we benchmark against other human-implemented experts). While we consider autonomous driving to be one useful application demonstrating the capabilities of LangProp, we wish to keep the implementation as simple as possible so that we can fairly assess the performance of the learning paradigm while keeping other things mostly standard. It would be interesting to see future work that integrates a trust framework into the system (e.g. by changing the training objective, or encouraging modeling of the actors’ trustworthiness in the prompt).
>
> 10. How does the quality of the dataset influence the performance of the LangProp because this can depend on what driving scenarios are considered?
>
> Yes, the quality and diversity of the dataset are important because that is how the model is trained and its performance assessed, similarly to most machine learning methods. As for the robustness and generalizability of our trained model, in fact our LangProp method is more generalizable and robust compared to neural models, because our LangProp “model” is implemented as logic and code, which tend to have more generalizability and can also take advantage of some cross-domain knowledge held by a general LLM, whereas neural models tend to require many training samples to learn a high-resolution input-output mapping. (For instance, learning a mapping of y = exp(cos(x)) may require many samples for a neural approach, whereas our code approach only has to generate a line of code corresponding to this function, hence it is much more sample-efficient.)

---

> ### Author Response · Authors · 2023-11-15
> **Authors' response [conclusion]**
>
> Overall, we appreciate the questions you have raised, which suggest some interesting directions for follow-up research on LangProp. However, we believe the research contribution, novelty, and the non-trivial amount of work (see our open-sourced code) haven’t been granted the recognition and fair assessment they deserve, and would like to request a reconsideration of the criticism and rating. Thank you very much for your contribution to this discussion.

---

> ### Author Response · Authors · 2023-11-22
> **Authors' response [Updated the paper reflecting your feedback]**
>
> Thank you again for your review. We found your feedback regarding safety concerns to be helpful, and have expanded our discussion on future work in Appendix E, where we stress the importance of safe deployment with additional risk mitigation mechanisms.
>
> We have also added additional examples featuring Sudoku and CartPole to the repository. Please refer to Appendix A.9, as well as https://github.com/langprop-iclr24/LangProp/tree/main/src/langprop#more-examples for the details on the experiments to show the generality of the framework. We would like to highlight that both experiments required no modifications to the pre-existing code-base, and that the training code for the new experiments is kept minimal and can be applied to other tasks. We have also added examples of checkpoints, prompts, initially obtained solutions and final solutions after training with LangProp. In particular, we would like to point you to the CartPole example, where the initial policy (https://github.com/langprop-iclr24/LangProp/blob/main/src/langprop/examples/cartpole/example_checkpoint/zero_shot_policy.txt) learned a simplistic policy and only obtained a score of 9.9, but the trained policy (https://github.com/langprop-iclr24/LangProp/blob/main/src/langprop/examples/cartpole/example_checkpoint/trained_policy.txt) refined its strategy to include a PID controller, obtaining a score of 500 (maximum).
>
> We acknowledge that some more complex scenarios may require further investigation, possibly requiring a hybrid approach of both an interpretable code-based policy and a neural network-based value estimator. This is also discussed in Appendix E, where we outline the characteristics of the current LangProp framework and how this may be improved.
>
> Thank you very much for helping us strengthen the paper, hopefully with more transparency.

---

### Author Response · Authors · 2023-11-22
**Authors' reponse [thanking reviewers and highlighting improvements made, including additional application examples]**

We would like to thank the reviewers for their input and for constructive suggestions.
We would like to highlight some of the discussions we had, and the improvements we have made to our submission reflecting the feedback we have received.

Firstly, we would like to re-emphasize the novelty of our LangProp approach as a learning paradigm that enables data-driven code optimization for problems that were not easily learnable with other machine learning methods (e.g. planning, logic, algorithms). We show LangProp’s capability of improving code compared to the LLM’s zero-shot performance, as well as comparing against human experts for the driving policy. As reviewers correctly point out, there are advantages of using neural networks that are not entirely replaceable by LangProp. We do not claim that a solution using LangProp is appropriate for all problems - in fact, neural networks excel in working with continuous state-action spaces and low-level control, whereas LLMs are better at handling high-level planning or reasoning tasks, rather than low-level control tasks. Our intention for this paper is to propose an alternative learning paradigm that allows LLMs to be used to learn high-level planning which has hitherto been a difficult problem for other machine learning approaches.

Another concern raised was on robustness and safety of code policies obtained by LangProp. We consider this a valid concern, and emphasize the importance of further research in our future work section. Note that many of the criticisms are not unique to LangProp but applies to LLMs in general, and LangProp is a framework that strictly improves the performance over zero-shot uses of LLMs. Further work is of course necessary, but we believe LangProp to be a step in the right direction in using LLMs safely.

The main request from multiple reviewers was to include additional experiments featuring applications of LangProp outside the domain of autonomous driving in CARLA to show generalizability. We have added additional examples featuring Sudoku and CartPole to the repository. Please refer to Appendix A.9, as well as https://github.com/langprop-iclr24/LangProp/tree/main/src/langprop#more-examples for the details on the experiments to show the generality of the framework. We would like to highlight that both experiments required no modifications to the pre-existing code-base, and that the training code for the new experiments is kept minimal and can be applied to other tasks. We have also added examples of checkpoints, prompts, initially obtained solutions and final solutions after training with LangProp.

We expanded our discussion in the future work section in Appendix E, where we outline the characteristics of the current LangProp framework and how this may be improved. We thank all the reviewers for contributing to this discussion, making the paper more transparent and grounded.

---

### Author Response · Authors · 2023-11-23
**Added driving policy example in Appendix E**

We have now added a driving code example generated by LangProp in Appendix E, in addition to having it in the open-sourced repository. (Hence, the Future Work section is now in Appendix F). We thank reviewer maNz for making this suggestion and helping us improve the discoverability of the code, and we thank all reviewers for their constructive feedback, helping us improve the quality of our work.

---

### Meta-Review · Area_Chair_7WLa · 2023-12-07

**Metareview:**

In this paper, the authors proposed LangProp, an approach to optimize generated codes by iteratively improving them with LLMs and supervised/ reinforcement learning. The authors applied the approach to a specific domain for autonomous driving in CARLA.

The reviewers appreciated the novelty of this approach to automatically improving generated code (including improvement in high-level planning and transparency) and applying it to a challenging task for autonomous driving policies. However, considering all the reviewers’ feedback and follow-up discussions with authors, there are still some major concerns in the current work: (i) Rule-based policy generation approach and its performance as compared to learnable neural systems, especially in more complex driving systems in actual usage; (ii) Limited experimental results only on a very specific task for autonomous driving in CARLA (I do note the application to additional tasks like Sudoku and CartPole, but these results are a bit post hoc). I highly encourage the authors to review all the reviewers’  feedback and suggestions. One area the authors could investigate more is the experimental results to showcase LangProp in more diverse tasks through more variants of LLMs (including open-source/ small-size models).

**Justification For Why Not Higher Score:**

While the proposed idea is indeed innovative, the experimental results are quite limited to very specific application domains and the results are not very impressive as compared to related baselines.

**Justification For Why Not Lower Score:**

N/A

---

### Decision · Program_Chairs · 2024-01-16

Reject